# Block Broyden's Methods for Solving Nonlinear Equations

**Chengchang Liu**
Department of Computer Science & Engineering
The Chinese University of Hong Kong
7liuchengchang@gmail.com

**Cheng Chen**\*
Shanghai Key Laboratory of Trustworthy Computing
East China Normal University
chchen@sei.ecnu.edu.cn

**Luo Luo**
School of Data Science
Fudan University
luoluo@fudan.edu.cn

**John C.S. Lui**
Department of Computer Science & Engineering
The Chinese University of Hong Kong
cslui@cse.cuhk.edu.hk

## Abstract

This paper studies quasi-Newton methods for solving nonlinear equations. We propose block variants of both good and bad Broyden's methods, which enjoy explicit local superlinear convergence rates. Our block good Broyden's method has a faster condition-number-free convergence rate than existing Broyden's methods because it takes the advantage of multiple rank modification on Jacobian estimator. On the other hand, our block bad Broyden's method directly estimates the inverse of the Jacobian provably, which reduces the computational cost of the iteration. Our theoretical results provide some new insights on why good Broyden's method outperforms bad Broyden's method in most of the cases. The empirical results also demonstrate the superiority of our methods and validate our theoretical analysis.

## 1 Introduction

In this paper, we consider solving the following nonlinear equation systems:

$$\mathbf{F}(\mathbf{x}) = \mathbf{0}, \tag{1}$$

where $\mathbf{x} \in \mathbb{R}^d$, $\mathbf{F}(\mathbf{x}) \overset{\text{def}}{=} [F_1(\mathbf{x}), \cdots, F_d(\mathbf{x})]^\top : \mathbb{R}^d \to \mathbb{R}^d$ and each $F_i(\mathbf{x})$ is differentiable. Solving nonlinear equations is one of the most important problems in scientific computing [38]. It has various applications including machine learning [3, 4, 12, 15, 44], game theory [18, 39], economics [2] and control systems [5, 37].

Newton's method and its variants [16, 25, 26] such as the Gauss–Newton method [19, 38], the Levenberg–Marquart method [17, 28, 34, 36] and the trust region method [40, 53] are widely adopted to solve the systems of nonlinear equations. These methods usually enjoy fast local superlinear rates.

---

\*The corresponding author

37th Conference on Neural Information Processing Systems (NeurIPS 2023).

Newton's method takes iterates of form

$$\mathbf{x}_{t+1} = \mathbf{x}_t - (\mathbf{J}(\mathbf{x}_t))^{-1}\mathbf{F}(\mathbf{x}_t),$$

where $\mathbf{J}(\mathbf{x}) \in \mathbb{R}^{d \times d}$ is the Jacobian at $\mathbf{x}$. Since computing the inverse of the exact Jacobian matrix requires $\mathcal{O}(d^3)$ running time, Newton's method suffers from expensive computation especially when solving the large-scale nonlinear equations [20, 47, 54].

Quasi-Newton methods have been proposed for avoiding the heavy computational cost of Newton-type methods while preserving good local convergence behaviour [7–11, 13, 24, 31, 32, 41–43, 45, 51]. Among these quasi-Newton methods, the Broyden's methods [6], including the good and the bad schemes [1, 30, 35], are considered to be the most effective methods for solving nonlinear equations. The Broyden's good method[2] approximates the Jacobian $\mathbf{J}(\mathbf{x}_t)$ by an estimator $\mathbf{B}_t$ and updates the Jacobian estimator in each round as $\mathbf{B}_{t+1} = \mathbf{B}_t + \mathbf{\Delta}_t$. Here $\mathbf{\Delta}_t$ is a rank-1 updating matrix constructed by the curvature information. Broyden et al. [11], Kelley and Sachs [27] proved that the good Broyden's method can achieve asymptotic local superlinear rates.

The bad Broyden's method approximates the inverse of the Jacobian by $\mathbf{H}_t$ and updates the approximate matrix directly. Although the bad Broyden's method enjoys less computational cost than good Broyden's method in each iteration, it does not perform as well as the good method in most cases [1]. Lin et al. [29] show that both the good and bad Broyden's methods have superlinear rates of $\mathcal{O}((1/\sqrt{t})^t)$ and provide some insights on the difference between their empirical performance.

Ye et al. [50] proposed a new variant of good Broyden's method by conducting $\mathbf{\Delta}_t$ with a greedy or random strategy. Their method achieves a better explicit convergence rate of $\mathcal{O}((1 - 1/d)^{t(t-1)/4})$. However, it remains unknown whether this convergence rate can be further improved by leveraging block updates which increase the reuse rate of the data in cache and take advantage of parallel computing [14]. Gower and Richtárik [21] studied several random quasi-Newton updates including the Broyden's updates for approximating the inverse of matrices, but they only provide implicit linear rates for their methods. Liu et al. [33] established explicit convergence rates for several block quasi-Newton updates, but they focus on approximating positive definite matrices.

In this paper, we propose two random block Broyden's methods for solving nonlinear equations and provide their explicit superlinear convergence rates. We compare the theoretical results of proposed methods with existing Broyden's methods in Table 1 and summarize our contribution as follows:

- We provide explicit convergence rates for the block good Broyden's udpate and the block bad Broyden's update proposed by Gower and Richtárik [21]. Our results show that the block good Broyden's update can approximate a nonsingular matrix $\mathbf{A}$ with a linear rate of $(1 - k/d)^t$ which improves the previous rate of $(1 - 1/d)^t$ where $k \stackrel{\text{def}}{=} \text{rank}(\mathbf{\Delta}_t)$. We also show that the "bad" update can approximate the inverse matrix $\mathbf{A}^{-1}$ with an linear rate of $(1 - k/(d\hat{\kappa}^2))^t$ where $\hat{\kappa}$ is the condition number of $\mathbf{A}$. To the best of our knowledge, this is the first explicit convergence rate for the block bad Broyden's update.

- We propose the block good Broyden's method with convergence rate $\mathcal{O}((1 - k/d)^{t(t-1)/4})$ where $k$ is the rank of the updating matrix $\mathbf{\Delta}_t$. This rate reveals the advantage of block update and improves previous results. Our method also relaxes the initial conditions stated in Ye et al. [50].

- We propose the block bad Broyden's method with convergence rate $\mathcal{O}((1 - k/(4d\kappa^2))^{t(t-1)/4})$. We also study the initial conditions of two proposed block variants. Our analysis shows that bad Broyden's method is only suitable for the cases where the condition number of the Jacobian is small, while good Broyden's method performs well in most cases.

**Paper Organization** In Section 2, we introduce the notation and assumptions as the preliminaries of this paper. In Section 3, we introduce the block good or bad Broyden's updates for approximating the general matrix. In Section 4, we propose the block good or bad Broyden's methods with explicit local superlinear rates. In Section 5, we discuss the behavior difference of the good and bad methods. We validate our methods by numerical experiments in Section 6. Finally, we conclude our results in Section 7. All proofs are deferred to appendix.

---

[2]We use the names "good Broyden's method" and "bad Broyden's method" by following the previous literature [1, 10, 22, 35].

Table 1: We summarize the properties of Broyden's methods for solving the Nonlinear equations

| Methods | rank($\mathbf{\Delta}_t$) | Convergence Rate |
|---|---|---|
| Good/Bad Broyden's Method [1, 6, 29] | 1 | $\mathcal{O}\left(1/t^{t/2}\right)$ |
| Greedy/Randomized Good Broyden's Method [50] | 1 | $\mathcal{O}\big((1-1/d)^{t(t-1)/4}\big)$ |
| Block Good Broyden's Method Algorithm 1 | $k \in [d-1]$ | $\mathcal{O}\big((1-k/d)^{t(t-1)/4}\big)$ |
| Block Bad Broyden's Method Algorithm 2 | $k \in [d]$ | $\mathcal{O}\big((1-k/(4\kappa^2 d))^{t(t-1)/4}\big)$ |

## 2 Preliminaries

We let $[d] \overset{\text{def}}{=} \{1, 2 \cdots, d\}$. We use $\| \cdot \|_F$ to denote the Frobenius norm of a given matrix, $\| \cdot \|_2$ to denote the spectral norm of a vector and Euclidean norm of a matrix respectively. The standard basis for $\mathbb{R}^d$ is presented by $\{\mathbf{e}_1, \cdots, \mathbf{e}_d\}$ and $\mathbf{I}_d$ is the identity matrix. We denote the trace, the largest singular value, and the smallest singular value of a matrix by $\text{tr}(\cdot)$, $\sigma_{\min}(\cdot)$, and $\sigma_{\max}(\cdot)$ respectively.

We use $\mathbf{x}_*$ to denote the solution of the nonlinear equation (1) and $\mathbf{J}_*$ to denote the Jacobian matrix at $\mathbf{x}_*$, i.e., $\mathbf{J}_* \overset{\text{def}}{=} \mathbf{J}(\mathbf{x}_*)$. We let $\mu \overset{\text{def}}{=} \sigma_{\min}(\mathbf{J}(\mathbf{x}_*))$, $L \overset{\text{def}}{=} \sigma_{\max}(\mathbf{J}(\mathbf{x}_*))$ and then define the condition number of $\mathbf{J}_*$ as $\kappa \overset{\text{def}}{=} L/\mu$. We also use $\hat{\kappa} \overset{\text{def}}{=} \sigma_{\max}(\mathbf{A})/\sigma_{\min}(\mathbf{A})$ to present the condition number of given matrix $\mathbf{A}$.

Then we present two standard assumptions on the nonlinear equations (1), which is widely used in previous works [16, 29, 50].

**Assumption 2.1.** The solution $\mathbf{x}_*$ of the nonlinear equation (1) is unique and nondegenerate, i.e.,

$$\mu \overset{\text{def}}{=} \sigma_{\min}(\mathbf{J}_*) > 0.$$

**Assumption 2.2.** The Jacobian $\mathbf{J}(\mathbf{x})$ satisfies

$$\|\mathbf{J}(\mathbf{x}) - \mathbf{J}_*\|_2 \leq M\|\mathbf{x} - \mathbf{x}_*\|_2 \quad \text{for all} \quad \mathbf{x} \in \mathbb{R}^d. \tag{2}$$

The following proposition shows that if $\mathbf{x}$ is in some local region of $\mathbf{x}_*$, the Jacobian matrix $\mathbf{J}(\mathbf{x})$ has a bounded condition number.

**Proposition 2.3.** *Suppose Assumptions 2.1 and 2.2 hold. For all $\mathbf{x}$ satisfies $\|\mathbf{x} - \mathbf{x}_*\|_2 \leq \mu^2/(6LM)$, we have*

$$\sigma_{\min}(\mathbf{J}(\mathbf{x})) \geq \frac{\mu}{\sqrt{2}} \quad \text{and} \quad \sigma_{\max}(\mathbf{J}(\mathbf{x})) \leq \sqrt{2}L.$$

We present two notations for the block Broyden's Update.

**Definition 2.4** (Block Good Broyden's Update)**.** Let $\mathbf{A}, \mathbf{B} \in \mathbb{R}^{d \times d}$. For any full column rank matrix $\mathbf{U} \in \mathbb{R}^{d \times k}$, we define

$$\text{Block-G-Broyden}(\mathbf{B}, \mathbf{A}, \mathbf{U}) \triangleq \mathbf{B} + (\mathbf{A} - \mathbf{B})\mathbf{U}\left(\mathbf{U}^\top\mathbf{U}\right)^{-1}\mathbf{U}^\top. \tag{3}$$

**Definition 2.5** (Block Bad Broyden's Update)**.** Let $\mathbf{A}, \mathbf{H} \in \mathbb{R}^{d \times d}$. For any full column rank matrix $\mathbf{U} \in \mathbb{R}^{d \times k}$, we define

$$\text{Block-B-Broyden}(\mathbf{H}, \mathbf{A}, \mathbf{U}) \triangleq \mathbf{H} + (\mathbf{I}_d - \mathbf{H}\mathbf{A})\mathbf{U}(\mathbf{U}^\top\mathbf{A}^\top\mathbf{A}\mathbf{U})^{-1}\mathbf{U}^\top\mathbf{A}^\top. \tag{4}$$

## 3 The Block Broyden's Updates for Approximating Matrices

In this section, we provide the linear convergence rates of the block good and bad Broyden's updates for approximating matrices. The theoretical results is summarized in Table 2.

Table 2: We summarize the properties of Broyden's updates for approximating a given nonsingular matrix $\mathbf{A}$ or $\mathbf{A}^{-1}$.

| Updates | Previous Results | Improved Results Theorem 3.1/3.2 | Measure |
|---|---|---|---|
| Block Good Broyden's Update | $\left(1 - \frac{1}{d}\right)^t$ [21, 50] [a] | $\left(1 - \frac{k}{d}\right)^t$ | $\mathbb{E}\left[\|(\mathbf{B}_t - \mathbf{A})\|_F^2\right]$ |
| Block Bad Broyden's Update | $(1 - \rho)^t$ [21] [b] | $\left(1 - \frac{k}{d\hat{\kappa}^2}\right)^t$ | $\mathbb{E}\left[\|(\mathbf{H}_t - \mathbf{A}^{-1})\|_F^2\right]$ |

(a). the result holds for $k = 1$ and it is unknown when $k > 1$.

(b). Gower and Richtárik [21] only prove that $\rho \in [0, \frac{k}{d}]$, but do not provide the explicit value of $\rho$.

The block good Broyden's update, which aims to compute an approximation of matrix $\mathbf{A}$, can be written as:

$$\mathbf{B}_{t+1} = \text{Block-G-Broyden}(\mathbf{B}_t, \mathbf{A}, \mathbf{U}_t).$$

The following theorem presents a linear convergence rate of $(1 - k/d)^t$ which is better than the rate $(1 - 1/d)^t$ provided by Gower and Richtárik [21], Ye et al. [50].

**Theorem 3.1.** *Assume that $\mathbf{A} \in \mathbb{R}^{d \times d}$ and $\mathbf{B}_0 \in \mathbb{R}^{d \times d}$. If we select $\mathbf{U}_t = [\mathbf{e}_{i_1}, \mathbf{e}_{i_2}, \cdots, \mathbf{e}_{i_k}] \in \mathbb{R}^{d \times k}$, where $\{i_1, \cdots, i_k\}$ are uniformly chosen from $\{1, 2, \cdots, d\}$ without replacement at each round, then for any nonsingular matrix $\mathbf{C} \in \mathbb{R}^{d \times d}$, the block good Broyden's update satisfies*

$$\|\mathbf{C}(\mathbf{B}_{t+1} - \mathbf{A})\|_F^2 \le \|\mathbf{C}(\mathbf{B}_t - \mathbf{A})\|_F^2, \tag{5}$$

*and*

$$\mathbb{E}\left[\|\mathbf{C}(\mathbf{B}_t - \mathbf{A})\|_F^2\right] \le \left(1 - \frac{k}{d}\right)^t \|\mathbf{C}(\mathbf{B}_0 - \mathbf{A})\|_F^2. \tag{6}$$

On the other hand, the bad Broyden's update which targets to approximate $\mathbf{A}^{-1}$ can be written as:

$$\mathbf{H}_{t+1} = \text{Block-B-Broyden}(\mathbf{H}_t, \mathbf{A}, \mathbf{U}_t).$$

Gower and Richtárik [21] provide an implicit rate of $(1 - \rho)^t$ for the above scheme with $\rho \in [0, k/d]$, but their analysis cannot guarantee an explicit $\rho$. In the following theorem, we show that the block bad Broyden's update can approximate $\mathbf{H}_t$ to $\mathbf{A}^{-1}$ with an explicit linear rate of $(1 - k/(\hat{\kappa}^2 d))^t$.

**Theorem 3.2.** *Assume that $\mathbf{A} \in \mathbb{R}^{d \times d}$ and $\mathbf{H}_0 \in \mathbb{R}^{d \times d}$. If we select $\mathbf{U}_t = [\mathbf{e}_{i_1}, \mathbf{e}_{i_2}, \cdots, \mathbf{e}_{i_k}] \in \mathbb{R}^{d \times k}$ where $\{i_1, \cdots, i_k\}$ are uniformly chosen from $\{1, 2, \cdots, d\}$ without replacement at each round, then for any nonsingular matrix $\mathbf{C} \in \mathbb{R}^{d \times d}$, the block bad Broyden's update satisfies*

$$\|\mathbf{C}(\mathbf{H}_{t+1} - \mathbf{A}^{-1})\|_F^2 \le \|\mathbf{C}(\mathbf{H}_t - \mathbf{A}^{-1})\|_F^2, \tag{7}$$

*and*

$$\mathbb{E}\left[\|\mathbf{C}(\mathbf{H}_t - \mathbf{A}^{-1})\|_F^2\right] \le \left(1 - \frac{k}{d\hat{\kappa}^2}\right)^t \|\mathbf{C}(\mathbf{H}_0 - \mathbf{A}^{-1})\|_F^2. \tag{8}$$

*Remark* 3.3. If we choose $\mathbf{C} = \mathbf{I}_d$ in Theorem 3.1 and Theorem 3.2, then the measures in these two theorems are exactly the same as the one in Section 8.5 and Section 8.3 of [21]. Besides, the rate of Theorem 3.1 recovers the convergent rates of Section 8.5 in [21] and Lemma 4.1 in [50] when we take $k = 1$.

## 4 The Block Broyden's Methods

In this section, we propose two block Broyden's methods for solving the nonlinear equation (1). We present our algorithms in section 4.1 and the corresponding convergence results in Section 4.2.

---

**Algorithm 1** Block Good Broyden's Method (BGB)

1: **Input:** Initial estimator $\mathbf{B}_0$, initial point $\mathbf{x}_0$ and block size $k$.

2: **for** $t = 0, 1 \ldots$

3:     $\mathbf{x}_{t+1} = \mathbf{x}_t - \mathbf{B}_t^{-1}\mathbf{F}(\mathbf{x}_t)$.

4:     Choose $\{i_1, i_2, \cdots, i_k\}$ by uniformly select $k$ items from $\{1, \cdots, d\}$ without replacement.

5:     $\mathbf{U}_t = [\mathbf{e}_{i_1}, \cdots, \mathbf{e}_{i_k}] \in \mathbb{R}^{d \times k}$.

6:     $\mathbf{B}_{t+1} = \text{Block-G-Broyden}(\mathbf{B}_t, \mathbf{J}(\mathbf{x}_{t+1}), \mathbf{U}_t)$.

7: **end for**

---

**Algorithm 2** Block Bad Broyden's Method (BBB)

1: **Input:** Initial estimator $\mathbf{H}_0$, initial point $\mathbf{x}_0$ and block size $k$.

2: **for** $t = 0, 1 \ldots$

3:     $\mathbf{x}_{t+1} = \mathbf{x}_t - \mathbf{H}_t\mathbf{F}(\mathbf{x}_t)$.

4:     Choose $\{i_1, \cdots, i_k\}$ by uniformly select $k$ items from $\{1, \cdots, d\}$ without replacement.

5:     $\mathbf{U}_t = [\mathbf{e}_{i_1}, \cdots, \mathbf{e}_{i_k}] \in \mathbb{R}^{d \times k}$.

6:     $\mathbf{H}_{t+1} = \text{Block-B-Broyden}(\mathbf{H}_t, \mathbf{J}(\mathbf{x}_{t+1}), \mathbf{U}_t)$.

7: **end for**

---

### 4.1 Algorithms

By using the block Broyden's updates in Section 3, we propose two novel algorithms called Block Good Broyden's Method (BGB) and Block Bad Broyden's Method (BBB) for solving nonlinear equations.

We present the BGB algorithm in Algorithm 1 which updates the Jacobian estimator $\mathbf{B}_t$ by the block good Broyden's update in each iteration. Notice that the inverse of $\mathbf{B}_t$ can be computed efficiently by adopting Sherman-Morrison-Woodbury formula [46]. On the other hand, the BBB algorithm, which is presented in Algorithm 2, approximates the inverse of the Jacobian directly by using the block bad Broyden's update. It usually has a lower computational cost than the BGB algorithm in each round because the BBB algorithm does not need to compute the inverse of the estimator $\mathbf{H}_t$.

*Remark* 4.1. Algorithms 1 and 2 do not require full information of the Jacobian. We construct $\mathbf{U_t}$ by subsampling the columns of the identity matrix. When updating the Jacobian estimator by the block updates, we need to calculate $\mathbf{J}_{t+1}\mathbf{U}_t$ which is only the partial information of $\mathbf{J}_{t+1}$ (columns of $\mathbf{J}_{t+1}$). Since we have $k \ll d$, it is not expensive to access the partial information of the Jacobian.

### 4.2 Convergence Analysis for the Block Broyden's Methods

We provide the convergence analysis for Algorithm 1 and Algorithm 2 in Section 4.2.1 and Section 4.2.2 respectively. We denote the Jabcobian matrix at $\mathbf{x}_t$ as $\mathbf{J}_t$. As previous works [16, 29, 50], we make an assumption on the estimator matrices in Algorithm 1 and Algorithm 2 as follows:

**Assumption 4.2.** We assume the sequence $\{\mathbf{B}_t\}_{t=0}^{\infty}$ generated by Algorithm 1 (and $\{\mathbf{H}_t\}_{t=0}^{\infty}$ generated by Algorithm 2) are well-defined and nonsingular.

#### 4.2.1 Analysis for Block Good Broyden's Methods

In this subsection, we use the following measures for our convergence analysis,

$$r_t \overset{\text{def}}{=} \|\mathbf{x}_t - \mathbf{x}_*\|_2 \quad \text{and} \quad \sigma_t \overset{\text{def}}{=} \|\mathbf{J}_*^{-1}(\mathbf{B}_t - \mathbf{J}_*)\|_F.$$

The $r_t$ measures the distance between $\mathbf{x}_t$ and the solution $\mathbf{x}_*$ and $\sigma_t$ measures how well does the estimator matrix $\mathbf{B}_t$ approximate the Jacobian at $\mathbf{x}_*$.

The following lemma provides upper bound of $\sigma_t$ after one block Broyden's update.

**Lemma 4.3.** *Performing Algorithm 1 under Assumptions 2.1, 2.2 and 4.2, we have*

$$\sigma_{t+1} \leq \sigma_t + \frac{2M\sqrt{d}}{\mu} r_{t+1} \qquad and \qquad \mathbb{E}[\sigma_{t+1}] \leq \sqrt{1 - \frac{k}{d}} \cdot \sigma_t + \frac{2M\sqrt{d}}{\mu} \cdot r_{t+1}. \qquad (9)$$

Based on Lemma 4.3, we present the superlinear convergence rate for Algorithm 1.

**Theorem 4.4.** *Suppose Assumptions 2.1, 2.2 and 4.2 hold and the initial condition of Algorithm 1 satisfies*

$$\frac{2M\sqrt{d}r_0}{\mu} \leq \min\left\{ \frac{(1-q)(d-k)}{4(1+q)d}, \frac{q}{4(1+q)} \right\} \quad and \quad \sigma_0 \leq \frac{q}{2(1+q)} \qquad (10)$$

*for arbitary $q \in (0, 1)$. Then for any $k \in [d-1]$, the output of Algorithm 1 satisfies*

$$\mathbb{E}\left[ \|\mathbf{J}_*^{-1}(\mathbf{B}_t - \mathbf{J}_*)\|_F \right] \leq 2\mathrm{e}\left(1 - \frac{k}{d}\right)^{t/2},$$

*and*

$$\mathbb{E}\left[ \frac{\|\mathbf{x}_{t+1} - \mathbf{x}_*\|_2}{\|\mathbf{x}_t - \mathbf{x}_*\|_2} \right] \leq 4\mathrm{e}\left(1 - \frac{k}{d}\right)^{t/2}.$$

Theorem 4.4 implies the following high probability bound for Algorithm 1.

**Corollary 4.5.** *Performing Algorithm 1 under the same assumption and initial condition as Theorem 4.4, with probability at least $1 - \delta$, we have*

$$\|\mathbf{J}_*^{-1}(\mathbf{B}_t - \mathbf{J}_*)\|_F \leq \frac{4\mathrm{e}d^2}{k^2\delta}\left(1 - \frac{k}{d+k}\right)^{t/2}, \qquad (11)$$

*and*

$$\|\mathbf{x}_t - \mathbf{x}_*\|_2 \leq \left(\frac{8\mathrm{e}d^2}{k^2\delta}\right)^t \left(1 - \frac{k}{d+k}\right)^{t(t-1)/4} \|\mathbf{x}_0 - \mathbf{x}_*\|_2. \qquad (12)$$

**Comparison with [50]** Compare Theorem 4.4 with Theorem 4.3 of [50], we can find that the convergence rate of our BGB algorithm is better than greedy and randomized good Broyden's methods [50] if we choose $k > 1$.

On the other hand, the initial condition of greedy and randomized good Broyden's methods [50] is

$$\|\mathbf{x}_0 - \mathbf{x}_*\|_2 = \mathcal{O}\left(\frac{\mu}{M\sqrt{d}}\right) \quad and \quad \|\mathbf{B}_0 - \mathbf{J}_0\|_F = \mathcal{O}(\mu), \qquad (13)$$

while the condition of Theorem 4.4 can be reformulated as

$$\|\mathbf{x}_0 - \mathbf{x}_*\|_2 = \mathcal{O}\left(\frac{\mu}{M\sqrt{d}}\right) \quad and \quad \|\mathbf{J}_*^{-1}(\mathbf{B}_0 - \mathbf{J}_*)\|_F = \mathcal{O}(1). \qquad (14)$$

Since

$$\|\mathbf{J}_*^{-1}(\mathbf{B}_0 - \mathbf{J}_*)\|_F \leq \|\mathbf{J}_*^{-1}(\mathbf{J}_* - \mathbf{J}_0)\|_F + \|\mathbf{J}_*^{-1}(\mathbf{B}_0 - \mathbf{J}_0)\|_F$$

$$\leq \frac{M\sqrt{d}}{\mu}\|\mathbf{x}_0 - \mathbf{x}_*\|_2 + \frac{1}{\mu}\|\mathbf{B}_0 - \mathbf{J}_0\|_F = \mathcal{O}(1),$$

condition (13) can implies condition (14). However, the reverse is not always true. For example, we can choose $\mathbf{B}_0 = 1.5\mathbf{J}_*$ and suppose

$$\mathbf{J}_0 = \mathbf{J}_* = \begin{bmatrix} 3 & 0 \\ 0 & 10^{-10} \end{bmatrix}.$$

Then we have $\|\mathbf{J}_*^{-1}(\mathbf{B}_0 - \mathbf{J}_*)\|_F = \|\frac{1}{2}\mathbf{I}_2\|_F = \mathcal{O}(1)$ while $\|\mathbf{B}_0 - \mathbf{J}_0\|_F = \|\frac{1}{2}\mathbf{J}_*\|_F \gg 10^{-10} = \mu$.

Overall, compared with the greedy or randomized good Broyden's method [50], Theorem 4.4 not only gives a faster convergence superlinear rate by leveraging the idea of block update, but also weakens the initial condition by using different measures in the analysis.

#### 4.2.2 Analysis for Block Bad Broyden's Methods

This subsection gives the convergence analysis for Algorithm 2. We use the following measures to describe the convergent behavior

$$R_t \overset{\text{def}}{=} \|\mathbf{J}_*(\mathbf{x}_t - \mathbf{x}_*)\|_2 \qquad \text{and} \qquad \tau_t \overset{\text{def}}{=} \|\mathbf{J}_*(\mathbf{H}_t - \mathbf{J}_*^{-1})\|_F.$$

The $R_t$ measures the distance between $\mathbf{x}_t$ and the solution $x_*$ and $\tau_t$ measures how well does the estimator $\mathbf{H}_t$ approximate the matrix $\mathbf{J}_*^{-1}$.

Using the convergence results for the block bad Broyden's update in Theorem 3.2, we are able to tackle the difference between the estimator $\mathbf{H}_t$ and the matrix $\mathbf{J}_*^{-1}$ after one block update in Algorithm 2.

**Lemma 4.6.** *Performing Algorithm 2 under Assumptions 2.1, 2.2 and 4.2 and suppose the sequence $\{\mathbf{x}_t\}_{t=0}^\infty$ generated by Algorithm 2 satisfies that $\|\mathbf{x}_t - \mathbf{x}_*\|_2 \le \mu^2/(6LM)$, we have*

$$\tau_{t+1} \le \tau_t + \frac{4M\sqrt{d}}{\mu^2} \cdot R_{t+1}^2 \quad \text{and} \quad \mathbb{E}[\tau_{t+1}] \le \sqrt{1 - \frac{k}{4\kappa^2 d}} \cdot \tau_t + \frac{4M\sqrt{d}}{\mu^2} \cdot R_{t+1}. \qquad (15)$$

We can establish the superlinear convergence of the block bad Broyden's method based on Lemma 4.6.

**Theorem 4.7.** *Suppose Assumptions 2.1, 2.2 and 4.2 hold and the initial condition of Algorithm 2 satisfies*

$$\frac{4M\sqrt{d}R_0}{\mu^2} \le \min\left\{\frac{1-q}{4}, \frac{q}{2}, \frac{\sqrt{d}}{3\kappa}\right\} \quad \text{and} \quad \tau_0 \le \frac{q}{2} \qquad (16)$$

*for arbitrary $q \in (0, 1)$. Then for $k \in [d]$, the output of Algorithm 2 satisfies*

$$\mathbb{E}\left[\|\mathbf{J}_*(\mathbf{H}_t - \mathbf{J}_*^{-1})\|_F\right] \le \mathrm{e}\left(1 - \frac{k}{4d\kappa^2}\right)^{t/2},$$

*and*

$$\mathbb{E}\left[\frac{\|\mathbf{J}_*(\mathbf{x}_{t+1} - \mathbf{x}_*)\|_2}{\|\mathbf{J}_*(\mathbf{x}_t - \mathbf{x}_*)\|_2}\right] \le 2\mathrm{e}\left(1 - \frac{k}{4d\kappa^2}\right)^{t/2}.$$

Similar to Corollary 4.5, we can also obtain the high probability bound for Algorithm 2.

**Corollary 4.8.** *Performing Algorithm 2 under the same assumption and initial condition as Theorem 4.7, with probability at least $1 - \delta$, we have*

$$\|\mathbf{J}_*(\mathbf{H}_t - \mathbf{J}_*^{-1})\|_F \le \frac{8\mathrm{e}d^2\kappa^4}{\delta k^2}\left(1 - \frac{k}{4d\kappa^2 + k}\right)^{t/2}, \qquad (17)$$

*and*

$$\|\mathbf{J}_*(\mathbf{x}_t - \mathbf{x}_*)\|_2 \le \left(\frac{16\mathrm{e}d^2\kappa^4}{k^2\delta}\right)^t \left(1 - \frac{k}{4d\kappa^2 + k}\right)^{t(t-1)/4} \|\mathbf{J}_*(\mathbf{x}_0 - \mathbf{x}_*)\|_2. \qquad (18)$$

## 5  Discussion

In this section, we discuss the performance difference between the good and bad Broyden's methods which is considered as an important open problem in the field of nonlinear equations [35].

We first discuss the different performance of the block Broyden's methods (Algorithm 1 and 2). Notice that the "good" method enjoys a condition-number-free superlinear rate of $\mathcal{O}((1-k/d)^{t(t-1)/4})$ and the initial conditions of $\mathbf{B}_0$ and $\mathbf{x}_0$ are $\|\mathbf{J}_*^{-1}(\mathbf{B}_0 - \mathbf{J}_*)\|_F = \mathcal{O}(1)$ and $\|\mathbf{x}_0 - \mathbf{x}_*\|_2 = \mathcal{O}\left(\frac{\mu}{M\sqrt{d}}\right)$ respectively. On the other hand, both the superlinear rate $\mathcal{O}((1 - k/(4d\kappa^2))^{t(t-1)/4})$ and initial conditions $\|\mathbf{J}_*(\mathbf{H}_0 - \mathbf{J}_*^{-1})\|_F = \mathcal{O}(\min\{1, \sqrt{d}/\kappa\})$, $\|\mathbf{J}_*(\mathbf{x}_0 - \mathbf{x}_*)\|_2 = \mathcal{O}(\mu^2/(M\sqrt{d}))$ for $\mathbf{H}_0$, $\mathbf{x}_0$ of the "bad" method depend on $\kappa$ heavily. Thus we think these two block Broyden's methods are suitable for different scenarios:

Table 3: Comparison between block good and bad Broyden's methods where $r_0 = \|\mathbf{x}_0 - \mathbf{x}_*\|_2$, $\sigma_0 = \|\mathbf{J}_*(\mathbf{B}_0 - \mathbf{J}_*)\|_F$, $R_0 = \|\mathbf{J}_*(\mathbf{x}_0 - \mathbf{x}_*)\|_2$ and $\tau_0 = \|\mathbf{J}_*(\mathbf{H}_0 - \mathbf{J}_*^{-1})\|_F$.

| | **Block Good Broyden's Method** Algorithm 1 | **Block Bad Broyden's Method** Algorithm 2 |
|---|---|---|
| Initial Condition | $\frac{M\sqrt{d}r_0}{\mu} = \mathcal{O}(1)$, $\sigma_0 = \mathcal{O}(1)$ | $\frac{M\sqrt{d}R_0}{\mu^2} = \mathcal{O}(1)$, $\tau_0 = \mathcal{O}(1 \wedge \frac{\sqrt{d}}{\kappa})$ |
| Superlinear Rate | $\mathcal{O}\left(\left(1 - \frac{k}{d}\right)^{t(t-1)/4}\right)$ | $\mathcal{O}\left(\left(1 - \frac{k}{4d\kappa^2}\right)^{t(t-1)/4}\right)$ |
| Suitable Scene | $\kappa \gg 1$ | $\kappa = \mathcal{O}(1)$ |

- The "good" method is more suitable for the cases of large condition number ($\kappa \gg 1$) because its convergence rate is condition-number-free and its initial condition has weaker dependency on $\kappa$ than the "bad" method.

- The 'bad' method may have better performance when $\kappa = \mathcal{O}(1)$ because under this case the convergence rates do not differ much between the "good" and "bad" method while the latter one usually has a cheaper computational cost per iteration.

The condition number is very large in most of the cases which means the "good" method generally outperforms the "bad" one. We summarize the different convergence rates, initial conditions and suitable scenes of the block good and bad Broyden's methods in Table 3.

The similar phenomenon also holds for the classical good and bad Broyden's methods [29], whose iterations can be reformulated as

$$\begin{cases} \mathbf{x}_{t+1} = \mathbf{x}_t - \mathbf{B}_t^{-1}\mathbf{F}(\mathbf{x}_t), \\ \mathbf{B}_{t+1} = \text{Block-G-Broyden}\left(\mathbf{B}_t, \hat{\mathbf{J}}_{t+1}, \mathbf{u}_t\right) = \mathbf{B}_t + \frac{(\mathbf{y}_t - \mathbf{B}_t\mathbf{u}_t)\mathbf{u}_t^\top}{\mathbf{u}_t^\top \mathbf{u}_t} \end{cases}, \tag{19}$$

and

$$\begin{cases} \mathbf{x}_{t+1} = \mathbf{x}_t - \mathbf{H}_t\mathbf{F}(\mathbf{x}_t), \\ \mathbf{H}_{t+1} = \text{Block-B-Broyden}\left(\mathbf{H}_t, \hat{\mathbf{J}}_{t+1}, \mathbf{u}_t\right) = \mathbf{H}_t + \frac{(\mathbf{u}_t - \mathbf{H}_t\mathbf{y}_t)\mathbf{y}_t^\top}{\mathbf{y}_t^\top \mathbf{y}_t} \end{cases} \tag{20}$$

respectively, where $\mathbf{u}_t = \mathbf{x}_{t+1} - \mathbf{x}_t$, $\hat{\mathbf{J}}_{t+1} = \int_0^1 \mathbf{J}(\mathbf{x}_t + s\mathbf{u}_t)\mathrm{d}s$ and $\mathbf{y}_t = \mathbf{F}(\mathbf{x}_{t+1}) - \mathbf{F}(\mathbf{x}_t)$. The different convergent behavior of the block Broyden's updates helps us understand the performance difference between the classical good and bad Broyden's methods for the similarity of their frameworks.

## 6 Experiments

We validate our methods on the Chandrasekhar H-equation which is well studied in the previous literature [25, 29, 50] as follows

$$F_i(\mathbf{x}) = x_i - \left(1 - \frac{c}{2N}\sum_{j=1}^{N}\frac{\mu_i x_j}{\mu_i + \mu_j}\right)^{-1}, \tag{21}$$

where $\mathbf{x} = [x_1, \cdots, x_N]^\top \in \mathbb{R}^N$ and $\mathbf{F}(\mathbf{x}) = [F_1(\mathbf{x}), \cdots, F_N(\mathbf{x})]^\top \in \mathbb{R}^N$. We denote GB-Cl and BB-Cl as the classical good and bad Broyden's methods respectively [1, 29]. We denote GB-Gr and GB-Ra as the greedy and randomized Broyden's methods [50] respectively. Our experiments are conducted on a PC with Apple M1 and all algorithms are implemented in Python 3.8.12.

Our first experiment considers three cases: $N = 200$, $N = 300$, $N = 400$. We set $c = 1 - 10^{-12}$ for the H-equation and choose the block size $k = N/10$ for the proposed methods. In all cases, we use the same inputs $\mathbf{B}_0 = 0.1\mathbf{I}_N$ ($\mathbf{H}_0 = 10\mathbf{I}_N$) for all algorithms. We use classical Newton method as the warm-up algorithm to obtain $\mathbf{x}_0$ which satisfies the local condition and take it as the

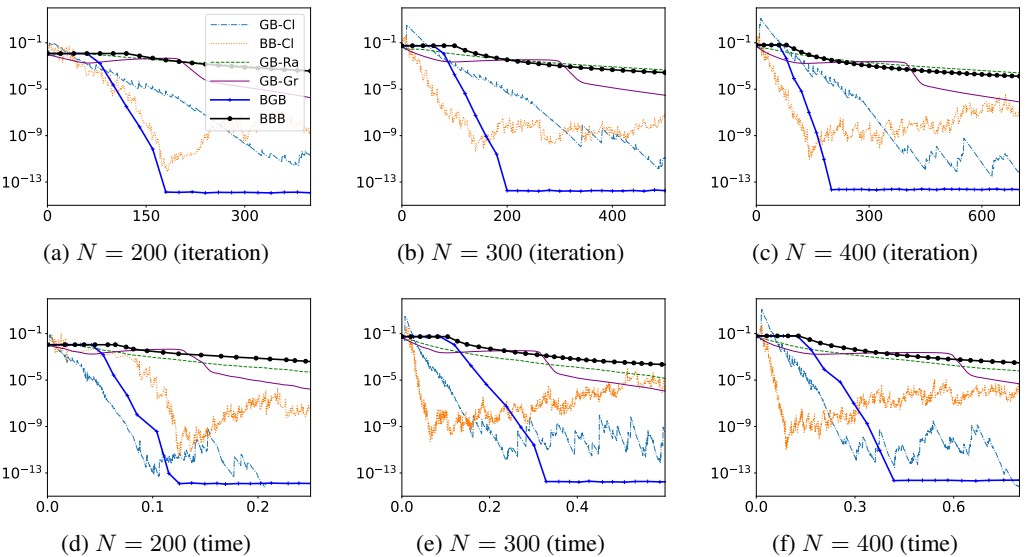

Figure 1: We demonstrate iteration numbers vs. $\|\mathbf{F}(\mathbf{x})\|_2$ and CPU time (second) vs. $\|\mathbf{F}(\mathbf{x})\|_2$ for H-equation with different equation numbers $N$.

initial point for all methods. We compare the proposed BGB and BBB algorithm with baselines and present the results of iteration number against $\|\mathbf{F}(\mathbf{x})\|_2$ and running time against $\|\mathbf{F}(\mathbf{x})\|_2$ in Figure 1. We observe that the proposed block good Broyden's method (BGB) outperforms the baselines in all cases, but the block bad Broyden's method (BBB) does not perform very well. This is mainly because $\kappa$ is very large in this setting ($\kappa \approx 10^6$). We also note that the classical Broyden's methods (GB-Cl and BB-Cl) are numerical unstable. Specifically, they do not guarantee the descent of $\|\mathbf{F}(\mathbf{x}_t)\|_2$ and encounter `nan` value during the iterations. The BB-Cl algorithm even fails to converge after some iterations. Such instability of the classical Broyden's methods is also observed in the previous literature [29, 50].

Our second experiment explores the performance of the proposed block Broyden's methods with different block size. We also study whether BBB algorithm has good performance for the nonlinear equation which Jacobian of the solution small condition number. By fixing $N = 400$ and setting $c = \{1-10^{-1}, 1-10^{-3}, 1-10^{-5}\}$, we obtain different condition numbers of (21) as $\kappa = 2, 31, 327$. We present the results in Figure 2. For each $\kappa$, we also vary the block size $k = \{1, 10, 100\}$ for BGB and BBB algorithms. We observe that when $\kappa = \mathcal{O}(1)$, BBB outperforms BGB in terms of the CPU time (Figure 2 (d), (e)). which matches our analysis in section 5. We also find that larger block size $k$ will lead to faster convergence in terms of the iterations ((a), (b), (c) of Figure 2), which verifies our theoretical results in section 4.2.

# 7 Conclusion

In this paper, we have proposed the block Broyden's methods for solving nonlinear equations. The proposed block good Broyden's method enjoys a faster superlinear rate than all of the existing Broyden's methods. We have also shown that the block bad Broyden's update approximates the inverse of the object matrix with an explicit linear rate and proposed the block bad Broyden's method accordingly. The established convergence results for the block good and bad methods bring us new understanding on the performance difference between the good and bad Broyden's methods. Especially, they can explain why good Broyden's method generally outperforms the "bad" one.

For the future work, it is possible to incorporate the safeguard mechanism in Wang et al. [48] to remove the assumption on the Jacobian estimator (Assumption 4.2). It will also be interesting to study the global behavior based on the recent advance in Jiang et al. [23] and design efficient stochastic or sketched algorithms [48, 49, 52] for solving nonlinear equations.

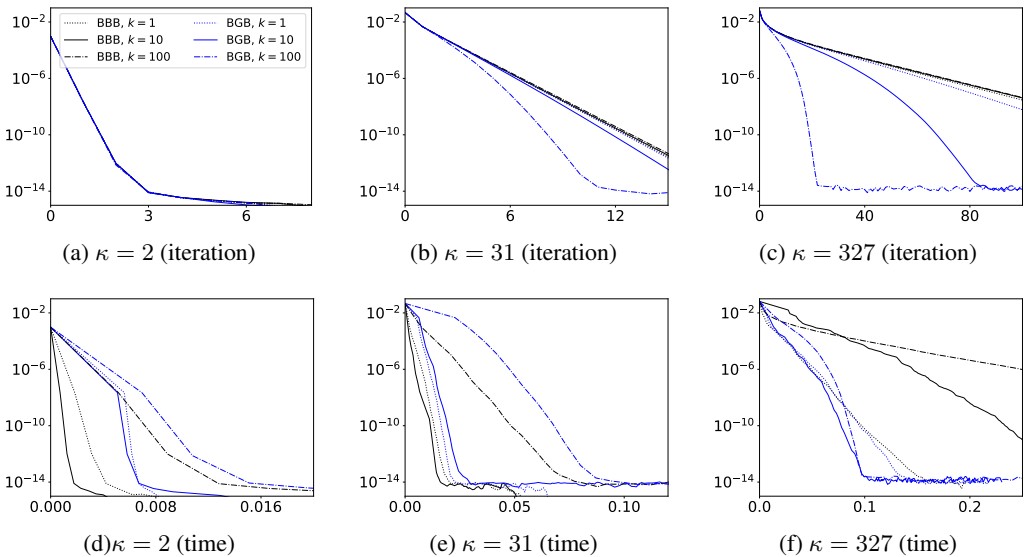

Figure 2: We demonstrate iteration numbers vs. $\|\mathbf{F}(\mathbf{x})\|_2$ and CPU time (second) vs. $\|\mathbf{F}(\mathbf{x})\|_2$ for H-equation with different condition number $\kappa$.

## Acknowledgement

We would like to thank Haishan Ye for valuable discussion. Cheng Chen is supported by National Natural Science Foundation of China (No. 62306116) and the Dean's fund of Shanghai Key Laboratory of Trustworthy Computing. Luo Luo is supported by National Natural Science Foundation of China (No. 62206058) and Shanghai Sailing Program (22YF1402900). John C.S. Lui is supported in part by the Hong Kong GRC 14215722.

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
