We present several useful lemmas in Section A. We give the detailed proof of Section 2 and 3 in Section B and C. The detailed proof of Broyden's good method (Section 4.2.1) and Broyden's bad method (Section 4.2.2) are presented in Section D and E respectively.

## A   Useful Lemmas

**Lemma A.1.** *Let* $\mathbf{U} = [\mathbf{e}_{i_1}, \mathbf{e}_{i_2}, \cdots, \mathbf{e}_{i_k}] \in \mathbb{R}^{d \times k}$ *where* $\{i_1, \cdots, i_k\}$ *are uniformly chosen from* $\{1, 2, \cdots, d\}$ *without replacement, then it holds that*

$$\mathbb{E}\left[\mathbf{U}(\mathbf{U}^\top \mathbf{U})^{-1} \mathbf{U}^\top\right] = \frac{k}{d} \mathbf{I}_d. \tag{22}$$

*Proof.* Since $\mathbf{U} = [\mathbf{e}_{i_1}, \mathbf{e}_{i_2}, \cdots, \mathbf{e}_{i_k}]$ and $i_1, \cdots, i_k$ are different, it always holds that $\mathbf{U}^\top \mathbf{U} = \mathbf{I}_k$, which means

$$\mathbb{E}[\mathbf{U}(\mathbf{U}^\top \mathbf{U})^{-1} \mathbf{U}^\top] = \mathbb{E}[\mathbf{U}\mathbf{U}^\top] = \mathbb{E}\left[\sum_{m=1}^{k} \mathbf{e}_{i_m} \mathbf{e}_{i_m}^\top\right] = \sum_{m=1}^{k} \mathbb{E}[\mathbf{e}_{i_m} \mathbf{e}_{i_m}^\top] = \frac{k}{d} \mathbf{I}_d.$$

$\square$

**Lemma A.2** ([17, Theorem 3.13]). *Let* $\mathbf{A}, \mathbf{B} \in \mathbb{R}^{d \times d}$, *then it holds that*

$$\|\mathbf{A}\mathbf{B}\|_F \leq \|\mathbf{A}\|_2 \|\mathbf{B}\|_F. \tag{23}$$

**Lemma A.3** (Modified from [32, Lemma 26], [52, Theorem 4.5]). *Suppose the nonnegative random sequences* $\{X_t\}$ *satisfies* $\mathbb{E}[X_t] \leq a (1 - 1/\eta)^{t/2}$ *and* $X_t \geq 0$ *for all* $t \geq 0$ *and some constants* $a \geq 0$ *and* $\eta > 1$. *Then for any* $\delta \in (0, 1)$, *we have*

$$X_t \leq \frac{2a\eta^2}{\delta} \left(1 - \frac{1}{1+\eta}\right)^{t/2} \tag{24}$$

*for all* $t$ *with probability at least* $1 - \delta$.

*Proof.* According to Markov's inequality, we have

$$\mathbb{P}\left(X_t \geq \frac{a}{\epsilon_t} \left(1 - \frac{1}{\eta}\right)^{t/2}\right) \leq \frac{\mathbb{E}[X_t]}{\frac{a}{\epsilon_t}\left(1 - \frac{1}{\eta}\right)^{t/2}} \leq \epsilon_t,$$

choose $\epsilon_t = \delta(1-s)s^t$ where $0 < s < 1$, we have

$$\mathbb{P}\left(X_t \geq \frac{a}{\epsilon_t} \left(1 - \frac{1}{\eta}\right)^{t/2}, \exists t \in \mathbb{N}\right) \leq \sum_{t=1}^{\infty} \delta(1-s)s^t = \delta.$$

With probability $1 - \delta$, we have

$$X_t \leq \left(\frac{1 - \frac{1}{\eta}}{s^2}\right)^{t/2} \cdot \frac{a}{(1-s)\delta},$$

for all $t$. Set $s = \sqrt{1 - 1/\eta^2} \leq 1 - 1/(2\eta^2)$, we obtain the result of (24). $\square$

## B   The Proof of Section 2

### B.1   The Proof of Proposition 2.3

*Proof.* Since $\|\mathbf{J}_*\|_2 \leq L$, it holds that

$$\|\mathbf{J}(\mathbf{x})\|_2 \leq \|\mathbf{J}(\mathbf{x}) - \mathbf{J}_*\|_2 + \|\mathbf{J}_*\|_2 \overset{(2)}{\leq} M\|\mathbf{x} - \mathbf{x}_*\|_2 + L \leq \sqrt{2}L.$$

It also holds that

$$\mathbf{J}_*^\top \mathbf{J}_* = \mathbf{J}(\mathbf{x})^\top \mathbf{J}(\mathbf{x}) + (\mathbf{J}_*^\top \mathbf{J}_* - \mathbf{J}(\mathbf{x})^\top \mathbf{J}(\mathbf{x}))$$

$$\preceq \mathbf{J}(\mathbf{x})^\top \mathbf{J}(\mathbf{x}) + \|(\mathbf{J}(\mathbf{x})^\top \mathbf{J}(\mathbf{x}) - \mathbf{J}_*^\top \mathbf{J}_*)\|_2 \mathbf{I}_d$$

$$\preceq \mathbf{J}(\mathbf{x})^\top \mathbf{J}(\mathbf{x}) + \|\mathbf{J}(\mathbf{x})\|_2 \|\mathbf{J}(\mathbf{x}) - \mathbf{J}_*\|_2 \mathbf{I}_d + \|\mathbf{J}_*\|_2 \|\mathbf{J}(\mathbf{x}) - \mathbf{J}_*\|_2 \mathbf{I}_d$$

$$\overset{(2)}{\preceq} \mathbf{J}(\mathbf{x})^\top \mathbf{J}(\mathbf{x}) + 3LM\|\mathbf{x} - \mathbf{x}^*\|\mathbf{I}_d \preceq \mathbf{J}(\mathbf{x})^\top \mathbf{J}(\mathbf{x}) + \frac{\mu^2}{2}\mathbf{I}_d,$$

which implies that

$$\sigma_{\min}(\mathbf{J}(\mathbf{x})) \geq \frac{\mu}{\sqrt{2}}.$$

$\square$

## C   The Proof of Section 3

### C.1   The Proof of Theorem 3.1

*Proof.* We first consider one step of the block good Broyden's update

$$\mathbf{B}_+ = \text{Block-G-Broyden}(\mathbf{B}, \mathbf{A}, \mathbf{U}).$$

According to the block Broyden's update rule of (3), we have

$$\mathbf{C}(\mathbf{B}_+ - \mathbf{A}) = \mathbf{C}(\mathbf{B} - \mathbf{A}) + \mathbf{C}(\mathbf{A} - \mathbf{B})\mathbf{U}\left(\mathbf{U}^\top \mathbf{U}\right)^{-1}\mathbf{U}^\top \tag{25}$$

$$= \mathbf{C}(\mathbf{B} - \mathbf{A})\left(\mathbf{I}_d - \mathbf{U}\left(\mathbf{U}^\top \mathbf{U}\right)^{-1}\mathbf{U}^\top\right). \tag{26}$$

Then it holds that

$$\begin{aligned}
&\mathbf{C}(\mathbf{B}_+ - \mathbf{A})(\mathbf{B}_+ - \mathbf{A})^\top \mathbf{C}^\top \\
&= \mathbf{C}(\mathbf{B} - \mathbf{A})\left(\mathbf{I}_d - \mathbf{U}\left(\mathbf{U}^\top \mathbf{U}\right)^{-1}\mathbf{U}^\top\right)\left(\mathbf{I}_d - \mathbf{U}\left(\mathbf{U}^\top \mathbf{U}\right)^{-1}\mathbf{U}^\top\right)(\mathbf{B} - \mathbf{A})^\top \mathbf{C}^\top \\
&= \mathbf{C}(\mathbf{B} - \mathbf{A})\left(\mathbf{I}_d - \mathbf{U}\left(\mathbf{U}^\top \mathbf{U}\right)^{-1}\mathbf{U}^\top\right)(\mathbf{B} - \mathbf{A})^\top \mathbf{C}^\top \\
&\preceq \mathbf{C}(\mathbf{B} - \mathbf{A})(\mathbf{B} - \mathbf{A})^\top \mathbf{C}^\top,
\end{aligned} \tag{27}$$

which proves (5).

According to Lemma A.1, we can obtain

$$\begin{aligned}
\mathbb{E}\left[\|\mathbf{C}(\mathbf{B}_+ - \mathbf{A})\|_F^2\right] &\overset{(25)}{=} \|\mathbf{C}(\mathbf{B} - \mathbf{A})\|_F^2 - \mathbb{E}\left[\text{tr}\left((\mathbf{B} - \mathbf{A})^\top \mathbf{C}^\top \mathbf{C}(\mathbf{B} - \mathbf{A})\mathbf{U}\left(\mathbf{U}^\top \mathbf{U}\right)^{-1}\mathbf{U}^\top\right)\right] \\
&= \|\mathbf{C}(\mathbf{B} - \mathbf{A})\|_F^2 - \text{tr}\left((\mathbf{B} - \mathbf{A})^\top \mathbf{C}^\top \mathbf{C}(\mathbf{B} - \mathbf{A})\mathbb{E}\left[\mathbf{U}\left(\mathbf{U}^\top \mathbf{U}\right)^{-1}\mathbf{U}^\top\right]\right) \\
&\overset{(22)}{=} \|\mathbf{C}(\mathbf{B} - \mathbf{A})\|_F^2 - \frac{k}{d}\|\mathbf{C}(\mathbf{B} - \mathbf{A})\|_F^2 \\
&= \left(1 - \frac{k}{d}\right)\|\mathbf{C}(\mathbf{B} - \mathbf{A})\|_F^2.
\end{aligned}$$

So we have

$$\mathbb{E}_t\left[\|\mathbf{C}(\mathbf{B}_{t+1} - \mathbf{A})\|_F^2\right] = \left(1 - \frac{k}{d}\right)\|\mathbf{C}(\mathbf{B}_t - \mathbf{A})\|_F^2.$$

Taking expectation on both sides of the above equation, we have

$$\mathbb{E}\left[\|\mathbf{C}(\mathbf{B}_{t+1} - \mathbf{A})\|_F^2\right] = \left(1 - \frac{k}{d}\right)\mathbb{E}[\|\mathbf{C}(\mathbf{B}_t - \mathbf{A})\|_F^2].$$

Thus, we obtain

$$\mathbb{E}\left[\|\mathbf{C}(\mathbf{B}_t - \mathbf{A})\|_F^2\right] = \left(1 - \frac{k}{d}\right)^t\|\mathbf{C}(\mathbf{B}_0 - \mathbf{A})\|_F^2.$$

$\square$

## C.2 The Proof of Theorem 3.2

*Proof.* We consider one step of the block bad Broyden's update

$$\mathbf{H}_+ = \text{Block-B-Broyden}(\mathbf{H}, \mathbf{A}, \mathbf{U}).$$

According to the update rule, it holds that

$$\mathbf{C}(\mathbf{H}_+ - \mathbf{A}^{-1}) = \mathbf{C}(\mathbf{H} - \mathbf{A}^{-1}) - \mathbf{C}(\mathbf{H} - \mathbf{A}^{-1})(\mathbf{A}\mathbf{U}(\mathbf{U}^\top\mathbf{A}^\top\mathbf{A}\mathbf{U})^{-1}\mathbf{U}^\top\mathbf{A}^\top)$$
$$= \mathbf{C}(\mathbf{H} - \mathbf{A}^{-1})(\mathbf{I}_d - \mathbf{A}\mathbf{U}(\mathbf{U}^\top\mathbf{A}^\top\mathbf{A}\mathbf{U})^{-1}\mathbf{U}^\top\mathbf{A}^\top),$$

which means

$$\mathbf{C}(\mathbf{H}_+ - \mathbf{A}^{-1})(\mathbf{H}_+ - \mathbf{A}^{-1})^\top\mathbf{C}^\top$$
$$= \mathbf{C}(\mathbf{H} - \mathbf{A}^{-1})(\mathbf{I}_d - \mathbf{A}\mathbf{U}(\mathbf{U}^\top\mathbf{A}^\top\mathbf{A}\mathbf{U})^{-1}\mathbf{U}^\top\mathbf{A}^\top)(\mathbf{H} - \mathbf{A}^{-1})^\top\mathbf{C}^\top$$
$$\preceq \mathbf{C}(\mathbf{H} - \mathbf{A}^{-1})(\mathbf{H} - \mathbf{A}^{-1})^\top\mathbf{C}^\top$$

which proves (7). Besides, it holds that

$$\mathbf{C}(\mathbf{H}_+ - \mathbf{A}^{-1})(\mathbf{H}_+ - \mathbf{A}^{-1})^\top\mathbf{C}^\top$$
$$= (\mathbf{H} - \mathbf{A}^{-1})(\mathbf{H} - \mathbf{A}^{-1})^\top - (\mathbf{H} - \mathbf{A}^{-1})(\mathbf{A}\mathbf{U}(\mathbf{U}^\top\mathbf{A}^\top\mathbf{A}\mathbf{U})^{-1}\mathbf{U}^\top\mathbf{A}^\top)(\mathbf{H} - \mathbf{A}^{-1})^\top$$

Since $\hat{\mu} = \min\sigma(\mathbf{A})$ and $\hat{L} = \max\sigma(\mathbf{A})$, we have

$$\hat{\mu}^2\mathbf{I} \preceq \mathbf{A}^\top\mathbf{A} \preceq \hat{L}^2\mathbf{I}, \tag{28}$$

which means

$$\mathbf{C}(\mathbf{H} - \mathbf{A}^{-1})(\mathbf{A}\mathbf{U}(\mathbf{U}^\top\mathbf{A}^\top\mathbf{A}\mathbf{U})^{-1}\mathbf{U}^\top\mathbf{A}^\top)(\mathbf{H} - \mathbf{A}^{-1})^\top\mathbf{C}^\top$$
$$\overset{(28)}{\succeq} \frac{1}{\hat{L}^2}\mathbf{C}(\mathbf{H} - \mathbf{A}^{-1})(\mathbf{A}\mathbf{U}(\mathbf{U}^\top\mathbf{U})^{-1}\mathbf{U}^\top\mathbf{A}^\top)(\mathbf{H} - \mathbf{A}^{-1})^\top\mathbf{C}^\top. \tag{29}$$

Combining the above results, we have

$$\mathbb{E}\big[\|\mathbf{C}(\mathbf{H}_+ - \mathbf{A}^{-1})\|_F^2\big]$$
$$\overset{(29)}{\leq} \|\mathbf{C}(\mathbf{H} - \mathbf{A}^{-1})\|_F^2 - \frac{1}{\hat{L}^2}\mathbb{E}\big[\text{tr}\big(\mathbf{C}(\mathbf{H} - \mathbf{A}^{-1})(\mathbf{A}\mathbf{U}(\mathbf{U}^\top\mathbf{U})^{-1}\mathbf{U}^\top\mathbf{A}^\top)(\mathbf{H} - \mathbf{A}^{-1})^\top\big)\mathbf{C}^\top\big]$$
$$= \|\mathbf{C}(\mathbf{H} - \mathbf{A}^{-1})\|_F^2 - \frac{1}{\hat{L}^2}\text{tr}\big(\mathbb{E}\big[\mathbf{C}(\mathbf{H} - \mathbf{A}^{-1})(\mathbf{A}\mathbf{U}(\mathbf{U}^\top\mathbf{U})^{-1}\mathbf{U}^\top\mathbf{A}^\top)(\mathbf{H} - \mathbf{A}^{-1})^\top\mathbf{C}^\top\big]\big)$$
$$= \|\mathbf{H} - \mathbf{A}^{-1}\|_F^2 - \frac{1}{\hat{L}^2}\text{tr}\big((\mathbf{H} - \mathbf{A}^{-1})\mathbf{A}\mathbb{E}\big[\mathbf{U}(\mathbf{U}\mathbf{U}^\top)^{-1}\mathbf{U}^\top\big]\mathbf{A}^\top(\mathbf{H} - \mathbf{A}^{-1})^\top\big)$$
$$\overset{(3)}{=} \|\mathbf{C}(\mathbf{H} - \mathbf{A}^{-1})\|_F^2 - \frac{k}{\hat{L}^2 d}\text{tr}\big(\mathbf{C}(\mathbf{H} - \mathbf{A}^{-1})\mathbf{A}\mathbf{A}^\top(\mathbf{H} - \mathbf{A}^{-1})^\top\mathbf{C}^\top\big)$$
$$\overset{(28)}{\leq} \|\mathbf{C}(\mathbf{H} - \mathbf{A}^{-1})\|_F^2 - \frac{k\hat{\mu}^2}{d\hat{L}^2}\text{tr}\big(\mathbf{C}(\mathbf{H} - \mathbf{A}^{-1})(\mathbf{H} - \mathbf{A}^{-1})^\top\mathbf{C}^\top\big)$$
$$= \left(1 - \frac{k}{d\hat{\kappa}^2}\right)\|\mathbf{C}(\mathbf{H} - \mathbf{A}^{-1})\|_F^2.$$

So we have

$$\mathbb{E}_t\big[\|\mathbf{C}(\mathbf{H}_{t+1} - \mathbf{A}^{-1})\|_F^2\big] = \left(1 - \frac{k}{d\hat{\kappa}^2}\right)\|\mathbf{C}(\mathbf{H}_t - \mathbf{A}^{-1})\|_F^2.$$

Taking expectation on both sides of the above equation, we have

$$\mathbb{E}\big[\|\mathbf{C}(\mathbf{H}_{t+1} - \mathbf{A}^{-1})\|_F^2\big] = \left(1 - \frac{k}{d\hat{\kappa}^2}\right)\mathbb{E}\big[\|\mathbf{C}(\mathbf{H}_t - \mathbf{A}^{-1})\|_F^2\big].$$

Thus, we obtain

$$\mathbb{E}\big[\|\mathbf{C}(\mathbf{H}_t - \mathbf{A}^{-1})\|_F^2\big] = \left(1 - \frac{k}{d\hat{\kappa}^2}\right)^t\|\mathbf{C}(\mathbf{H}_0 - \mathbf{A}^{-1})\|_F^2.$$

$\square$

# D  The Proof of Section 4.2.1

## D.1  The Proof of Lemma 4.3

*Proof.* Take $\mathbf{C} = \mathbf{J}_*^{-1}$ in (5) and (6) of Theorem 3.1, we have

$$\mathbb{E}\left[\|\mathbf{J}_*^{-1}(\mathbf{B}_{t+1} - \mathbf{J}_{t+1})\|_F^2\right] \leq \left(1 - \frac{k}{d}\right)\|\mathbf{J}_*^{-1}(\mathbf{B}_t - \mathbf{J}_{t+1})\|_F^2, \tag{30}$$

and

$$\|\mathbf{J}_*^{-1}(\mathbf{B}_{t+1} - \mathbf{J}_{t+1})\|_F^2 \leq \|\mathbf{J}_*^{-1}(\mathbf{B}_t - \mathbf{J}_{t+1})\|_F^2. \tag{31}$$

We have

$$\begin{aligned}
\mathbb{E}\left[\sigma_{t+1}\right] &= \mathbb{E}\left[\|\mathbf{J}_*^{-1}(\mathbf{B}_{t+1} - \mathbf{J}_*)\|_F\right] \\
&\leq \mathbb{E}\left[\|\mathbf{J}_*^{-1}(\mathbf{B}_{t+1} - \mathbf{J}_{t+1})\|_F\right] + \|\mathbf{J}_*^{-1}(\mathbf{J}_{t+1} - \mathbf{J}_*)\|_F \\
&\overset{(30)}{\leq} \sqrt{1 - \frac{k}{d}} \cdot \|\mathbf{J}_*^{-1}(\mathbf{B}_t - \mathbf{J}_{t+1})\|_F + \|\mathbf{J}_*^{-1}(\mathbf{J}_{t+1} - \mathbf{J}_*)\|_F \\
&\leq \sqrt{1 - \frac{k}{d}} \cdot \|\mathbf{J}_*^{-1}(\mathbf{B}_t - \mathbf{J}_*)\|_F + 2\|\mathbf{J}_*^{-1}(\mathbf{J}_{t+1} - \mathbf{J}_*)\|_F \\
&\overset{(23)}{\leq} \sqrt{1 - \frac{k}{d}} \cdot \|\mathbf{J}_*^{-1}(\mathbf{B}_t - \mathbf{J}_*)\|_F + 2\|\mathbf{J}_*^{-1}\|_2\|\mathbf{J}_{t+1} - \mathbf{J}_*\|_F \\
&\overset{(2)}{\leq} \sqrt{1 - \frac{k}{d}} \cdot \sigma_t + \frac{2M\sqrt{d}}{\mu} \cdot \|\mathbf{x}_{t+1} - \mathbf{x}_*\|_2 \\
&= \sqrt{1 - \frac{k}{d}} \cdot \sigma_t + \frac{2M\sqrt{d}}{\mu} \cdot r_{t+1}.
\end{aligned}$$

Similarly, it holds that

$$\begin{aligned}
\sigma_{t+1} &= \|\mathbf{J}_*^{-1}(\mathbf{B}_{t+1} - \mathbf{J}_*)\|_F \\
&\leq \|\mathbf{J}_*^{-1}(\mathbf{B}_{t+1} - \mathbf{J}_{t+1})\|_F + \|\mathbf{J}_*^{-1}(\mathbf{J}_{t+1} - \mathbf{J}_*)\|_F \\
&\overset{(31)}{\leq} \|\mathbf{J}_*^{-1}(\mathbf{B}_t - \mathbf{J}_{t+1})\|_F + \|\mathbf{J}_*^{-1}(\mathbf{J}_{t+1} - \mathbf{J}_*)\|_F \\
&\leq \|\mathbf{J}_*^{-1}(\mathbf{B}_t - \mathbf{J}_*)\|_F + 2\|\mathbf{J}_*^{-1}(\mathbf{J}_{t+1} - \mathbf{J}_*)\|_F \\
&\overset{(23)}{\leq} \|\mathbf{J}_*^{-1}(\mathbf{B}_t - \mathbf{J}_*)\|_F + 2\|\mathbf{J}_*^{-1}\|_2\|\mathbf{J}_{t+1} - \mathbf{J}_*\|_F \\
&\overset{(2)}{\leq} \sigma_t + \frac{2M\sqrt{d}}{\mu}\|\mathbf{x}_{t+1} - \mathbf{x}_*\|_2 \\
&= \sigma_t + \frac{2M\sqrt{d}}{\mu}r_{t+1}.
\end{aligned}$$

$\square$

## D.2  The Proof of Theorem 4.4

We first provide two useful lemmas

**Lemma D.1** ([31, Lemma 9]). *Under the same assumptions of Theorem 4.4, taking the iteration of* $\mathbf{x}_{t+1} = \mathbf{x}_t - \mathbf{B}_t^{-1}\mathbf{F}(\mathbf{x}_t)$ *and* $\mathbf{B}_t$ *is nonsingular, it holds that*

$$r_{t+1} \leq \|\mathbf{B}_t^{-1}\mathbf{J}_*\|_2 \left(\sigma_t r_t + \frac{M}{\mu}r_t^2\right),$$

*if* $\sigma_t \leq 1$, *we have*

$$r_{t+1} \leq \frac{1}{1 - \sigma_t}\left(\sigma_t r_t + \frac{2M}{\mu}r_t^2\right). \tag{32}$$

**Lemma D.2.** *Under the same assumptions of Theorem 4.4 with the following initial conditions*

$$2Mr_0\sqrt{d}/\mu \le \min\left\{\frac{1-q}{4(q+1)}, \frac{q}{4(q+1)}\right\} \quad and \quad \sigma_0 \le \frac{q}{2(1+q)}, \tag{33}$$

*we have*

$$r_{t+1} \le qr_t.$$

*Proof.* We use induction to prove the following facts that

$$r_{t+1} \le qr_t, \tag{34}$$

and

$$\sigma_t \le \sigma_0 + \frac{q}{1-q} \cdot \frac{2M\sqrt{d}}{\mu} \cdot r_0 \le \frac{3q}{4(1+q)}. \tag{35}$$

For $t = 0$, we have $\sigma_0 \le 1$, it holds that

$$r_1 \overset{(32)}{\le} \frac{\sigma_0 + 2Mr_0/\mu}{1-\sigma_0} \cdot r_0 \overset{(33)}{\le} \frac{2q/(2(1+q))}{1-q/(2(1+q))} \le qr_0,$$

Suppose (34) and (35) hold for all $t = 0, \cdots t' - 1$, then for $t = t'$, we have

$$\sigma_{t'} \overset{(9)}{\le} \sigma_{t'-1} + \frac{2M\sqrt{d}}{\mu} \cdot r_{t'} \overset{(9)}{\le} \sigma_{t'-2} + \frac{2M\sqrt{d}}{\mu} \cdot r_{t'-1} + \frac{2M\sqrt{d}}{\mu} \cdot r_{t'}$$

$$\overset{(34)}{\le} \cdots \le \sigma_0 + \frac{2M\sqrt{d}}{\mu}\sum_{i=1}^{t'} r_i \le \sigma_0 + \frac{2M\sqrt{d}}{\mu}\sum_{i=1}^{t'} q^i r_0$$

$$\le \sigma_0 + \frac{q}{1-q}\frac{2M\sqrt{d}}{\mu}r_0 \le \frac{3q}{4(1+q)},$$

which means $\sigma_{t'} \le 1$, thus we have

$$r_{t'+1} \overset{(32)}{\le} \frac{1}{1 - 3q/(4(1+q))}\left(\frac{3q}{4(1+q)} + \frac{2M}{\mu}r_0\right)r_{t'}$$

$$\overset{(33)}{\le} \frac{4(1+q)}{4+q} \cdot \frac{4q}{4(1+q)}r_{t'} \le qr_{t'}.$$

$\square$

Now, we prove the results of Theorem 4.4

*Proof.* We denote $\alpha \overset{\text{def}}{=} \sqrt{1 - k/d}$. It holds that

$$\mathbb{E}_t[\sigma_{t+1}] \overset{(9)}{\le} \alpha\left(\sigma_t + \frac{2M\sqrt{d}}{\mu\alpha} \cdot r_t\right), \tag{36}$$

and according to Lemma D.2, we have

$$\frac{1}{1-\sigma_t} \overset{(35)}{\le} \frac{1}{1 - 3q/(4(1+q))} \le 1 + q,$$

which implies that

$$r_{t+1} \overset{(32)}{\le} \alpha\left(\sigma_t + \frac{2Mr_t}{\mu}\right)\left(\frac{1+q}{\alpha}\right)r_t \tag{37}$$

$$\le \alpha\left(\sigma_t + \frac{2M\sqrt{d}r_t}{\mu\alpha}\right)\left(\frac{1+q}{\alpha}\right)r_t. \tag{38}$$

We denote $\eta_t = \sigma_t + 2M\sqrt{d}r_t/(\mu\alpha)$, then it holds that

$$\mathbb{E}_t[\eta_{t+1}] \overset{(36),(37)}{\leq} \alpha\eta_t\left(1 + \frac{2M\sqrt{d}(1+q)}{\mu\alpha^2} \cdot r_t\right)$$

$$\leq \alpha\eta_t\exp\left(\frac{2M\sqrt{d}(1+q)}{\mu\alpha^2}r_t\right) \overset{(34)}{\leq} \alpha\eta_t\exp\left(\frac{2M\sqrt{d}(1+q)}{\mu\alpha^2}q^tr_0\right).$$

Taking expectation on both sides of the above inequality, we have

$$\mathbb{E}[\eta_{t+1}] \leq \alpha\exp\left(\frac{2M\sqrt{d}(1+q)}{\mu\alpha^2}q^tr_0\right)\mathbb{E}[\eta_t]$$

$$\leq \alpha^2\exp\left(\frac{2M\sqrt{d}(1+q)}{\mu\alpha^2}(q^t + q^{t-1})r_0\right)\mathbb{E}[\eta_{t-1}]$$

$$\cdots$$

$$\leq \alpha^{t+1}\exp\left(\frac{2M\sqrt{d}(1+q)}{\mu\alpha^2}\sum_{i=0}^{t}q^ir_0\right)\eta_0 \qquad (39)$$

$$\leq \alpha^{t+1}\exp\left(\frac{2M\sqrt{d}(1+q)}{\mu\alpha^2(1-q)}r_0\right)\eta_0$$

$$\leq \alpha^{t+1}2e,$$

where the last inequality comes from the initial condition that

$$r_0 \leq \frac{(1-q)\mu\alpha^2}{2(1+q)M\sqrt{d}},$$

and

$$\eta_0 = \sigma_0 + \frac{2M\sqrt{d}r_0}{\mu\alpha} \leq 1 + \frac{1}{2} \overset{(10)}{\leq} 2.$$

So, we have

$$\mathbb{E}\left[\frac{r_{t+1}}{r_t}\right] \overset{(37)}{\leq} \mathbb{E}[(1+q)\eta_t] \overset{(39)}{\leq} 4e\alpha^t,$$

and

$$\mathbb{E}[\sigma_t] \overset{(36)}{\leq} \mathbb{E}[\eta_t] \overset{(39)}{\leq} 2e\alpha^t.$$

$\square$

### D.3 The Poof of Corollary 4.5

*Proof.* We follow the notation and the results obtained in the proof of Theorem 4.4. Using Lemma A.3 with $a = 2e$, $\eta = d/k$ and $X_t = \sigma_t$, we obtain (11). Using Lemma A.3 with $a = 4e$, $\eta = d/k$ and $X_t = r_{t+1}/r_t$, we have

$$r_{t+1} \leq \frac{8ed^2}{k^2\delta}\left(1 - \frac{k}{d+k}\right)^{t/2}r_t,$$

holds for all $t$ with probability at least $1 - \delta$. by telescoping the above inequality, we can obtain (12). $\square$

## E The Proof of Section 4.2.2

In the following analysis, we denote

$$\hat{\mu} \overset{\text{def}}{=} \frac{\mu}{\sqrt{2}}, \qquad \hat{L} \overset{\text{def}}{=} \sqrt{2}L, \qquad \text{and} \qquad \hat{\kappa} \overset{\text{def}}{=} 2\kappa \qquad (40)$$

to simplify the presentation.

## E.1 The Proof of Lemma 4.6

*Proof.* Take $\mathbf{C} = \mathbf{J}_*$ in Theorem 3.2, we have

$$\mathbb{E}\left[\|\mathbf{J}_*(\mathbf{H}_{t+1} - \mathbf{J}_{t+1}^{-1})\|_F^2\right] \leq \left(1 - \frac{k}{d\hat{\kappa}^2}\right)\|\mathbf{J}_*(\mathbf{H}_t - \mathbf{J}_{t+1}^{-1})\|_F^2, \tag{41}$$

and

$$\|\mathbf{J}_*(\mathbf{H}_{t+1} - \mathbf{J}_{t+1}^{-1})\|_F^2 \leq \|\mathbf{J}_*(\mathbf{H}_t - \mathbf{J}_{t+1}^{-1})\|_F^2, \tag{42}$$

We have

$$\begin{aligned}
\mathbb{E}\left[\tau_{t+1}\right] &= \mathbb{E}\left[\|\mathbf{J}_*(\mathbf{H}_{t+1} - \mathbf{J}_*^{-1})\|_F\right] \\
&\leq \mathbb{E}\left[\|\mathbf{J}_*(\mathbf{H}_{t+1} - \mathbf{J}_{t+1}^{-1}) + \mathbf{J}_*(\mathbf{J}_{t+1}^{-1} - \mathbf{J}_*^{-1})\|_F\right] \\
&\stackrel{(41)}{\leq} \sqrt{1 - \frac{k}{d\hat{\kappa}^2}}\|\mathbf{J}_*(\mathbf{H}_t - \mathbf{J}_{t+1}^{-1})\|_F + \|\mathbf{J}_*(\mathbf{J}_{t+1}^{-1} - \mathbf{J}_*^{-1})\|_F \\
&\leq \sqrt{1 - \frac{k}{d\hat{\kappa}^2}} \cdot \|\mathbf{J}_*(\mathbf{H}_t - \mathbf{J}_*^{-1})\|_F + 2\|\mathbf{J}_*(\mathbf{J}_{t+1}^{-1} - \mathbf{J}_*^{-1})\|_F \\
&= \sqrt{1 - \frac{k}{d\hat{\kappa}^2}} \cdot \tau_t + 2\|\mathbf{J}_{t+1}^{-1}(\mathbf{J}_* - \mathbf{J}_{t+1})\|_F \\
&\stackrel{(23)}{\leq} \sqrt{1 - \frac{k}{d\hat{\kappa}^2}} \cdot \tau_t + 2\|\mathbf{J}_{t+1}^{-1}\|_2\|\mathbf{J}_* - \mathbf{J}_{t+1}\|_F \\
&\stackrel{(2)}{\leq} \sqrt{1 - \frac{k}{d\hat{\kappa}^2}} \cdot \tau_t + \frac{2M\sqrt{d}}{\hat{\mu}} \cdot r_{t+1} \\
&\leq \sqrt{1 - \frac{k}{d\hat{\kappa}^2}} \cdot \tau_t + \frac{2M\sqrt{d}}{\hat{\mu}^2} \cdot R_{t+1}.
\end{aligned}$$

Besides, it holds that

$$\begin{aligned}
\tau_{t+1} &\leq \|\mathbf{J}_*(\mathbf{H}_{t+1} - \mathbf{J}_{t+1}^{-1}) + \mathbf{J}_*(\mathbf{J}_{t+1}^{-1} - \mathbf{J}_*^{-1})\|_F \\
&\stackrel{(42)}{\leq} \|\mathbf{J}_*(\mathbf{H}_t - \mathbf{J}_{t+1}^{-1})\|_F + \|\mathbf{J}_*(\mathbf{J}_{t+1}^{-1} - \mathbf{J}_*^{-1})\|_F \\
&\leq \|\mathbf{J}_*(\mathbf{H}_t - \mathbf{J}_*^{-1})\|_F + 2\|\mathbf{J}_*(\mathbf{J}_{t+1}^{-1} - \mathbf{J}_*^{-1})\|_F \\
&= \tau_t + 2\|\mathbf{J}_{t+1}^{-1}(\mathbf{J}_* - \mathbf{J}_{t+1})\|_F \\
&\stackrel{(23)}{\leq} \tau_t + 2\|\mathbf{J}_{t+1}^{-1}\|_2\|\mathbf{J}_* - \mathbf{J}_{t+1}\|_F \\
&\stackrel{(2)}{\leq} \tau_t + \frac{2M\sqrt{d}}{\hat{\mu}} \cdot r_{t+1} \\
&\leq \tau_t + \frac{2M\sqrt{d}}{\hat{\mu}^2} \cdot R_{t+1} = \tau_t + \frac{4M\sqrt{d}}{\mu^2} \cdot R_{t+1}.
\end{aligned}$$

$\square$

## E.2 The Proof of Theorem 4.7

We first present two useful lemmas which use the same assumptions as Theorem 4.7 for the following analysis

**Lemma E.1** ([31, Lemma 11]). *Under the same assumptions of Theorem 4.7, taking the iteration of* $\mathbf{x}_{t+1} = \mathbf{x}_t - \mathbf{H}_t\mathbf{F}(\mathbf{x}_t)$*, it holds that*

$$R_{t+1} \leq \tau_t R_t + \frac{(1 + \tau_t)M}{2\mu^2} \cdot R_t^2. \tag{43}$$

**Lemma E.2.** *If* $\mathbf{x} \in \Omega^* \stackrel{def}{=} \{\mathbf{x} : \|\mathbf{J}_*(\mathbf{x} - \mathbf{x}_*)\|_2 \leq \mu^3/(6LM)\}$*, it holds* $\|\mathbf{x} - \mathbf{x}_*\| \leq \mu^2/(6LM)$*.*

*Proof.* We have

$$\|\mathbf{x} - \mathbf{x}_*\|_2 \le \|\mathbf{J}_*^{-1}\|_2 \|\mathbf{J}_*(\mathbf{x} - \mathbf{x}_*)\|_2 \le \frac{1}{\mu} \cdot \frac{\mu^3}{6LM} = \frac{\mu^2}{6LM}.$$

$\square$

**Lemma E.3.** *Under the same assumptions of Theorem 4.7 with the following initial conditions*

$$\frac{4M\sqrt{d}R_0}{\mu^2} \le \min\left\{\frac{1-q}{4}, \frac{q}{2}, \frac{\sqrt{d}}{3\kappa}\right\} \qquad and \qquad \tau_0 \le \frac{q}{2}, \tag{44}$$

*we have*

$$R_{t+1} \le qR_t \qquad and \qquad \|\mathbf{x}_t - \mathbf{x}^*\|_2 \le \frac{\mu^2}{6LM} \tag{45}$$

*hold for all $t$.*

*Proof.* We use induction to prove that

$$R_{t+1} \le qR_t \le q^{t+1}R_0, \tag{46}$$

and

$$\tau_t \le \frac{3q}{4}. \tag{47}$$

For $t = 0$, we have

$$R_1 \le \left(\tau_0 + \frac{M}{\hat{\mu}^2} \cdot R_0\right) R_0 \le qR_0,$$

Suppose (46) and (47) hold for $t = 0, \cdots t' - 1$, then we have

$$R_{t'} \le R_0 \le \frac{\mu^3}{6LM},$$

which means $\mathbf{x}_{t'} \in \Omega^*$. For $t = t'$, since $\mathbf{x}_0, \cdots, \mathbf{x}_{t'} \in \Omega^*$, using 2.3 and 4.6, it holds that

$$\tau_{t'} \overset{(15)}{\le} \tau_{t'-1} + \frac{2M\sqrt{d}}{\hat{\mu}^2} R_{t'} \overset{(15)}{\le} \tau_{t'-2} + \frac{2M\sqrt{d}}{\hat{\mu}^2} R_{t'-1} + \frac{2M\sqrt{d}}{\hat{\mu}^2} R_{t'}$$

$$\le \cdots \overset{(15)}{\le} \tau_0 + \frac{2M\sqrt{d}}{\hat{\mu}^2} \sum_{i=1}^{t'} R_i \overset{(46)}{\le} \tau_0 + \frac{2M}{\hat{\mu}^2} \sum_{i=1}^{t'} q^i R_0$$

$$\le \tau_0 + \frac{q}{1-q} \cdot \frac{2M\sqrt{d}}{\hat{\mu}^2} R_0 \overset{(44)}{\le} \frac{3q}{4},$$

and

$$R_{t'+1} \overset{(43)}{\le} \left(\tau_{t'} + \frac{M\sqrt{d}}{\hat{\mu}^2} \cdot R_{t'}\right) R_{t'} \overset{(46)}{\le} \left(\frac{3q}{4} + \frac{M\sqrt{d}}{\hat{\mu}^2} \cdot R_0\right) R_{t'} \overset{(44)}{=} qR_{t'},$$

which finish the induction. According to Lemma E.2 it holds for that

$$\|\mathbf{x}_t - \mathbf{x}_*\|_2 \le \frac{\mu^2}{6LM}.$$

$\square$

Now, we prove Theorem 4.7.

*Proof.* According to Lemma E.3, we always have $\|\mathbf{x}_t - \mathbf{x}_*\|_2 \le \mu^2/(6LM)$. Using Lemma 4.6 and Lemma E.1, we have

$$\mathbb{E}_t[\tau_{t+1}] \overset{(15)}{\le} \beta\left(\tau_t + \frac{4M\sqrt{d}}{\mu^2}\cdot R_t\right), \tag{48}$$

and

$$R_{t+1} \le \beta\left(\tau_t + \frac{4M\sqrt{d}}{\mu^2}\cdot R_t\right)2R_t, \tag{49}$$

where we assume that $\hat{\kappa} \ge \sqrt{2}$ and denote $\beta \overset{\text{def}}{=} \sqrt{1 - k/(d\kappa^2)}$, $\eta_t \overset{\text{def}}{=} \tau_t + 4M\sqrt{d}R_t/\mu^2$. We have

$$\mathbb{E}_t[\eta_{t+1}] \overset{(48),(49)}{\le} \beta\eta_t\left(1 + \frac{8M\sqrt{d}}{\mu^2}\cdot R_t\right)$$

$$\overset{(45)}{\le} \beta\eta_t\exp\left(\frac{8M\sqrt{d}\,q^tR_0}{\mu^2}\right),$$

taking expectation on both sides of the above inequality, we have

$$\mathbb{E}\left[\eta_{t+1}\right] \le \beta\exp\left(\frac{8M\sqrt{d}\,q^tR_0}{\mu^2}\right)\mathbb{E}\left[\eta_t\right]$$

$$\le \beta^2\exp\left(\frac{8M\sqrt{d}\,(q^t + q^{t-1})R_0}{\mu^2}\right)\mathbb{E}\left[\eta_{t-1}\right]$$

$$\cdots$$

$$\le \beta^{t+1}\exp\left(\frac{8M\sqrt{d}}{\mu^2}\sum_{i=0}^{t}q^iR_0\right)\eta_0 \tag{50}$$

$$\le \beta^{t+1}\exp\left(\frac{8\sqrt{d}M}{\mu^2(1-q)}R_0\right)\eta_0$$

$$\le \beta^{t+1}\mathrm{e},$$

where the last inequality comes from the initial condition that

$$R_0 \overset{(16)}{\le} \frac{(1-q)\mu^2}{8M\sqrt{d}},$$

and

$$\eta_0 = \tau_0 + \frac{4M\sqrt{d}R_0}{\mu^2} \overset{(16)}{\le} \frac{1}{2} + \frac{1}{2} \le 1.$$

So, we have

$$\mathbb{E}\left[\frac{R_{t+1}}{R_t}\right] \overset{(49)}{\le} \mathbb{E}\left[2\eta_t\right] \overset{(50)}{\le} 2\mathrm{e}\beta^t,$$

and

$$\mathbb{E}\left[\tau_t\right] \overset{(48)}{\le} \mathbb{E}\left[\eta_t\right] \overset{(50)}{\le} \mathrm{e}\beta^t.$$

$\square$

### E.3 The Proof of Corollary 4.8

*Proof.* We follow the notation and the results obtained in the proof of Theorem 4.7. Using Lemma A.3 with $a = \mathrm{e}$, $\eta = k/(4d\kappa^2)$ and $X_t = \tau_t$, we obtain (17). Using Lemma A.3 with $a = 2\mathrm{e}$, $\eta = k/(4d\kappa^2)$ and $X_t = R_{t+1}/R_t$, we have

$$R_{t+1} \le \left(\frac{32d^2\kappa^4\mathrm{e}}{k^2\delta}\right)\left(1 - \frac{k}{4d\kappa^2 + k}\right)^{t/2}R_t,$$

by telescoping the above inequality, we can obtain (18). $\square$

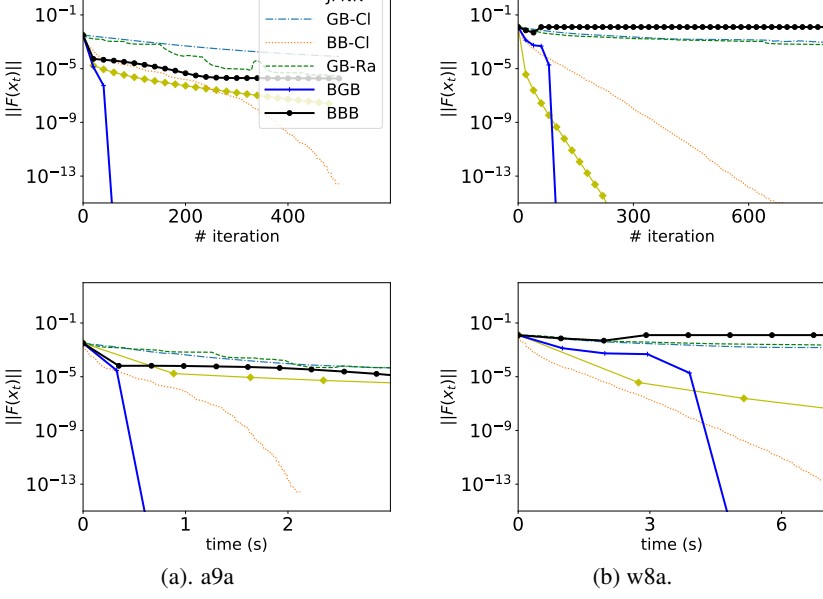

(a). a9a                                                  (b) w8a.

Figure 3: We demonstrate iteration numbers vs. $\|\mathbf{F}(\mathbf{x})\|$ and CPU time (second) vs. $\|\mathbf{F}(\mathbf{x})\|$ for solving logistic regression problem on real world datasets "a9a" and "w8a".

## F Additional Experiments

To verify the efficiency of our methods on real-world data, we adopt the proposed block Broyden's methods to solve the classical logistic regression:

$$\min_{\mathbf{x} \in \mathbb{R}^d} \frac{1}{n} \sum_{i=1}^{n} \ln(1 + \exp(-b_i \mathbf{a}_i^\top \mathbf{x})) + \frac{\lambda}{2} \|\mathbf{x}\|^2,$$

which corresponds to solving the following nonlinear equations

$$\lambda \mathbf{x} - \frac{1}{n} \sum_{i=1}^{n} \frac{\exp\left(b_i \mathbf{a}_i^\top \mathbf{x}\right)}{1 + \exp(-b_i \mathbf{a}_i \mathbf{x})} \cdot b_i \mathbf{a}_i = \mathbf{0}.$$

We compare the proposed methods BGB and BBB with GB-Cl, BB-Cl, GB-Ra and JFNK (Jacobian-Free Newton–Krylov) method [29]. We do not compare them with GB-Gr because it uses greedy strategy to choose $\mathbf{U}_t$, which requires to access the full Jacobian and thus is too expensive in practice. We set the initial Jacobian estimator $\mathbf{B}_0 = \mathbf{I}$ for all cases and validate our methods on two real world datasets "a9a" and "w8a" from the LIBSVM dataset [14] and present the results in Figure 3. The results demonstrate that the proposed BGB method outperforms the baselines significantly for the logistic regression.