# OpenReview forum: "Block Broyden's Methods for Solving Nonlinear Equations"
_NeurIPS.cc/2023/Conference — NeurIPS 2023 poster_

### Official Review · Reviewer_X3mb · 2023-07-04

**Soundness:** 3 good
**Presentation:** 3 good
**Contribution:** 3 good
**Rating:** 5
**Confidence:** 2

**Summary:**

In this work the authors introduce block variants of both good and bad Broyden's methods, which exhibit explicit local superlinear convergence rates. The block good Broyden's method, in particular, demonstrates a faster convergence rate compared to existing Broyden's methods that are not dependent on the condition number. This is achieved by leveraging multiple rank modifications on the Jacobian estimator. On the other hand, the block bad Broyden's method directly estimates the inverse of the Jacobian, resulting in reduced computational costs during the iteration process. The theoretical findings offer new insights into why the good Broyden's method tends to outperform the bad Broyden's method in most cases. Empirical results further validate the superiority of the proposed methods and affirm the theoretical analysis conducted by the authors.

**Strengths:**

* The authors provide explicit convergence rates for the block good Broyden’s update and the block bad Broyden’s update
* The block good Broyden’s update can approximate a nonsingular matrix A with a linear rate of $(1-k/d)^t$  and the “bad” update can approximate the inverse matrix $A^{−1}$ with an linear rate of $(1-k/(d \hat{\kappa}^2))^t$
* They propose the first explicit convergence rate for the block bad Broyden’s update.
* The assumptions are supported by theory and experiments.

**Weaknesses:**

In the experiment section Figure 1 and Figure 2 are a little bit confusing as for example Figure 1(a) and Figure 1(d) represents the experiments for the same N so they can be grouped under the same subfigure.

**Questions:**

* Can you compare the proposed method with the results presented by [1]?
*  Why did you choose Chandrasekhar H-equation to conduct the experiments for the given method?


[1] Robert M Gower and Peter Richtarik. Randomized quasi-newton updates are linearly convergent matrix inversion algorithms. arXiv preprint arXiv:1602.01768, 2016.

**Limitations:**

None.

---

> ### Author Rebuttal · Authors · 2023-08-09
>
> We thank the reviewer for the positive and helpful comments.
>
> > In the experiment section Figure 1 and Figure 2 are a little bit confusing as for example Figure 1(a) and Figure 1(d) represents the experiments for the same N so they can be grouped under the same subfigure.
>
> **Response:**
> We thank the author for pointing this and would like to improve the figures in the revision based on the suggestion.
>
>
> > Can you compare the proposed method with the results presented by [1]?
>
> **Response:**
> We make comprehensive comparison to the results presented by [1] according to the reviewer's suggestion. Please refer to Section 2 in the global response.
>
> > Why did you choose Chandrasekhar H-equation to conduct the experiments for the given method?
>
> **Response:**
> The Chandrasekhar H-equation plays an important role in scientific computing [A, B]. As shown in [C] and [Section 5.6, 20], its discrete version can be used in a wide class of problems of analytical radiative transfer theory. Thus we choose it to verify the empirical performance of our methods. In addition, recent studies [24, 38] on nonlinear equations also perform experiments on the H-equation.
>
>
>
>
> **References**
>
> [A] Subrahmanyan Chandrasekhar, Radiative Transfer, Dover, New York, 1960.
>
> [B] Richard W. Leggett. A new approach to the H-equation of Chandrasekhar. SIAM Journal on Mathematical Analysis, 7(4):542-550, 1976:
>
> [C] C. T. Kelley. Approximate methods for the solution of the Chandrasekhar H-equation. Journal of Mathematical Physics, 23(11):2097-2100, 1982.

---

> > ### Comment · Reviewer_X3mb · 2023-08-13
> >
> > I appreciate your responses and the explanation you provided for my questions. I have decided to keep my score as it is.

---

### Official Review · Reviewer_XomP · 2023-07-04

**Soundness:** 3 good
**Presentation:** 3 good
**Contribution:** 3 good
**Rating:** 6
**Confidence:** 4

**Summary:**

This paper extends the Broyden family quasi-Newton method into block setting and shows explicit local linear convergence rate under mild conditions. More specifically, the authors studied both the “good” and “bad” Broyden algorithms, namely the update on the Hessian/Jacobian and the inverse Hessian/Jacobian respectively. The authors provided some insights on why the “good” update is better than the “bad” update and provided numerical experiments to support their findings.

**Strengths:**

The theory is well-rounded and the method is consistent with existing works. The analysis on Algorithm 2 is interesting which incorporates the condition number of the Jacobian and brings insights on why the convergence of “bad” Broyden method is worse in practice.

**Weaknesses:**

(Please reply to the Questions section directly) First the “good” Broyden family still computes the inverse of matrix in the update; Second, the Assumption 4.1 is imposed toward the iterates directly, which is not very good; Third, it’s not very clear why we consider the block update. Also the numerical experiment is not adequate.

**Questions:**

1. Perhaps the biggest question I have toward the comparison of “good” and “bad” Broyden is that for the good Broyden method, still it computes the inverse of $B_{t}^{-1}$, which means that the dependency on the condition number is implicitly incorporated. I’d appreciate to hear from the author on how this problem is addressed (and how previous literature deals with this problem);
2. In assumption 4.1, the assumption is imposed toward the sequence $\{B_{t}\}$, which is not very good. Is there a possible safeguard mechanism so that the Jacobians are well-defined? This is pretty common in quasi-Newton literatures, such as [1].
3. Another problem is that the block update seems to be lack of motivations. Isn’t the case $k=1$ already enough to show the dependency of $\kappa$ for the “bad” Broyden update? Certainly the $k$ in each of the convergence result and the numerical experiments show the efficiency of block updates, but usually block updates are for bigger targets such as parallelization or decentralized update. Could the authors give some comment on this direction;
4. The numerical experiments on Chandrasekhar H-equation is not sufficient to show the efficiency of the proposed method. It would be interesting to see applications and experiments on problems with real-world data. In particular, would the “bad” Broyden method be better when the problem dimension is relatively large?


References:
[1] Wang, Xiao, et al. "Stochastic quasi-Newton methods for nonconvex stochastic optimization." SIAM Journal on Optimization 27.2 (2017): 927-956.

**Limitations:**

The limitation is well stated in weakness and question sections.

I’m not aware of any potential negative social impact of this work.

---

> ### Author Rebuttal · Authors · 2023-08-09
>
> We thank the reviewer for the positive and helpful comments.
>
> > Perhaps the biggest question I have toward the comparison of “good” and “bad” Broyden is that for the good Broyden method, still it computes the inverse of ${\bf B}_t^{-1}$, which means that the dependency on the condition number is implicitly incorporated. I’d appreciate to hear from the author on how this problem is addressed (and how previous literature deals with this problem);
>
> **Response:**
> Notice that in the analysis of the good Broyden's method, we only care about the error between the estimator ${\bf B}\_t$ and the Jacobian ${\bf J}\_\*$, i.e., $\|\|{\bf C}({\bf B}\_t-{\bf J}\_\*)\|\|\_F$, rather than the error between ${\bf B}\_t^{-1}$ and ${\bf J}\_\*^{-1}$. Thus the convergence rate of the good Broyden's method is condition-number free.
>
> Previous works [25], [38] also established such condition-number free superlinear rates for quasi-Newton methods on convex optimization and nonlinear equations respectively. Their analysis is also based on controlling the error between the estimator matrix and the exact Jacobian (or Hessian) matrix which do not incorporate the condition number.
>
>
> On the other hand, the analysis of the bad Broyden's method considers the error between ${\bf H}_t$ and ${\bf J}\_\*^{-1}$, which incorporates the condition number in the convergence rate.
>
>
>
> > In assumption 4.1, the assumption is imposed toward the sequence ${\bf B}_t$, which is not very good. Is there a possible safeguard mechanism so that the Jacobians are well-defined? This is pretty common in quasi-Newton literatures, such as [1].
>
> **Response:** We provide discussion on Assumption 4.1, illustrate that it is a reasonable assumption and give some potential ways to eliminate this Assumption. Please refer to the global response (Section 3).
>
> > Another problem is that the block update seems to be lack of motivations. Isn’t the case already enough to show the dependency of $\kappa$ for the “bad” Broyden update? Certainly the $k$ in each of the convergence result and the numerical experiments show the efficiency of block updates, but usually block updates are for bigger targets such as parallelization or decentralized update. Could the authors give some comment on this direction;
>
> **Response:**
> The block updates are very important for high performance computing. It can increase the reuse rate of the data in cache and take advantage of parallel computing. For more details, please refer to [A]. We will provide discussion on this topic in the revision.
>
>
> > The numerical experiments on Chandrasekhar H-equation is not sufficient to show the efficiency of the proposed method. It would be interesting to see applications and experiments on problems with real-world data. In particular, would the “bad” Broyden method be better when the problem dimension is relatively large?
>
> **Response:**
> We have added experiments based on the reviewer's suggestion, please refer to the global response (Section 1). We did not find that the block bad method would be better than the block good method when the dimension is relatively large.
>
>
> **References**
>
> [A] Tim Davis. Block matrix methods: Taking advantage of high-performance computers. Technical Report TR-98-024, Computer and Information Sciences Department, University of Florida, 1998.

---

> > ### Comment · Reviewer_XomP · 2023-08-12
> >
> > Thank you for your detailed responses to my comments and questions. I still believe that paper is useful contribution to the field and will keep my score.

---

### Official Review · Reviewer_YNqM · 2023-07-06

**Soundness:** 3 good
**Presentation:** 4 excellent
**Contribution:** 3 good
**Rating:** 7
**Confidence:** 4

**Summary:**

This paper studies block Broyden's methods for solving nonlinear equation systems and presents explicit local superlinear convergence rates. For the block good Broyden's method, the convergence rate is independent of the condition number of the Jacobian matrix at the solution, which that of the block bad Broyden's method depends on the condition number heavily. Numerical experiments validate the theoretical analysis.

**Strengths:**

- The authors provided explicit local superlinear convergence rates for  block good Broyden's method and block bad Broyden's method. The rate improves previous results and reveals the advantage of block update.

- The established convergence results give new understanding on the performance difference between the good and bad Broyden's methods.

- The paper is clearly written and well-organized.

**Weaknesses:**

- In Algotithm 1 and Algotithm 2, the Jacobian matrix is explicitly needed even when $k=1$, while that is not the case for classic Broyden's methods.
- The convergence rate depends on the dimension. When $d\gg 1$, the rate $1-1/d$  is close to $1$ and the convergence will be slow.
- Essentially the algorithms and analysis is the block version of previous work [38]. Although the comparison with [38] is include, it is still not so clear if this generalization (from rank $1$ to rank $k$ ) is straight-forward.

**Questions:**

- The rank $k$ is not defined in the contribution 1 when it is firstly used.

- What is the definition of $\hat{\kappa}$ in line 51, equation (8) and Table 2?

- I suggest to clarify the meaning of $\kappa$ in line 59.

- In line 78, $x^*$ is defined as 'the solution'. I suggest to clarify the uniqueness of the solution for the problem (1) considered. The nonlinear equation (1) may have multiple solutions.

- What is the meaning of the bracket notation, for example that used in Table 1?

- What is the meaning of $\operatorname{e}$'s in equation (11), (12), (17) and  (18)?

- Typos: line 24, 'large-scale'; it seems to be '$d$' (other than '$\operatorname{d}$') in  equation (11), (17) and (18).

- Typos: line 94 summarized **in** Table 2.

- Typos: line 150, our BGB algorithm is better **than** greedy ...

- Please check the format of the reference list, particularly the letter case (in [3], [5], [9], [13]) and the math symbol (in [15]), etc.

---

> ### Author Rebuttal · Authors · 2023-08-09
>
> We thank the reviewer for the positive and helpful comments.
>
> > In Algotithm 1 and Algotithm 2, the Jacobian matrix is explicitly needed even when $k=1$, while that is not the case for classic Broyden's methods.
>
> **Response:**
> Our algorithms **do not** require the full information of the Jacobian matrix. When updating the Jacobian estimator by the block updates (line 6 of Algorithm 1 and 2), we only need to calculate $k$ columns of the Jacobian matrix which is selected by the sampling matrix ${\bf U}_t$. Since $k\ll d$, it is cheap to obtain the partial information of the Jacobian. Using this partial information of the Jacobian can significantly improves the convergence rates compared to the classical Broyden's methods (see Table 1).
>
> The efficiency of proposed methods are also validated in our experiments part. As a result, although our methods need to calculate a bit more information per iteration, they require less running time than classical ones (see Figure 1 (d), (e), (f)).
>
>
>
>
> > The convergence rate depends on the dimension. When $d\gg 1$, the rate $1-1/d$ is close to 1
>  and the convergence will be slow.
>
> **Response:**
> Our block good Broyden's method has the rate of $\mathcal{O}((1-k/d)^{t(t-1)/4})$, while the classical Broyden's method converges with the rate of $\mathcal{O}(1/{t}^{t/2})$.
> Hence, our method is faster than the classical one if the dimension satisfies $d \le \mathcal{O}(kt/\ln(t))$.
>
> On the other hand, such dimension dependency rates are commonly appeared in greedy or random quasi-Newton methods, even for the convex minimization problems [A, 24].
>
>
> > Essentially the algorithms and analysis is the block version of previous work [38]. Although the comparison with [38] is include, it is still not so clear if this generalization (from rank 1 to rank $k$) is straight-forward.
>
> **Response:**
> The generalization from rank-$1$ to rank-$k$ is not straight-forward. None of previous work on block updates demonstrate the superiority of using rank-$k$ update over the rank-1 update. For example, [18] propose the block bad Broyden's update for matrix approximation but the its analysis only with the implicit linear rate ([Section 8.3, 18]), while our analysis (Theorem 3.2) clearly show the advance of using rank-$k$ update.
>
> In addition, our work does not only improve the convergence rates by using the block updates compared with [38], but also weaken the initial condition by carefully choosing the measure (line 135) and generalizing the lemma for the matrix approximation (Theorem 3.1 and Remark 3.3) where we allow any nonsingular matrix ${\bf C}$ in equation (5) and (6). We show that our initial condition is strictly weaker than that in [38] (line 149-157).
>
> Also, [38] only gives the random or greedy version of the good Broyden's methods, we present the block bad Broyden's methods which are novel and firstly show the estimator matrix converges to the ${\bf J}_*^{-1}$.
>
> > The rank $k$ is not defined in the contribution 1 when it is firstly used.
>
> **Response:**
> $k$ is the rank of the differential between the updated matrix and the original matrix in the matrix approximation update in the contribution 1. We will clarify this in the revision.
>
> > What is the definition of $\hat{\kappa}$ in line 51, equation (8) and Table 2?
>
> **Response:**
> In line 51, equation (8) and Table 2, we define $\hat{\kappa}=\sigma_{\max}({\bf A})/\sigma_{\min}({\bf A})$ where $\sigma_{\max}(\bf A)$, $\sigma_{\min}(\bf A)$ are the largest and smallest singular value of a given non-singular matrix ${\bf A}$ respectively.
>
> > I suggest to clarify the meaning of $\kappa$ in line 59.
>
> **Response:**  In line 59, we will define $\kappa$ as the condition-number of ${\bf J}({\bf x}^\*)$.
>
> > In line 78, ${\bf x}^*$ is defined as 'the solution'. I suggest to clarify the uniqueness of the solution for the problem (1) considered. The nonlinear equation (1) may have multiple solutions.
>
> **Response:** Thanks for the advice, we will clarify the uniqueness of the solution of eq.1.
>
> > What is the meaning of the bracket notation, for example that used in Table 1?
>
> **Response:** The notation $[d]$ means $\{1,2\cdots,d\}$.
>
> > What is the meaning of $e$'s in equation (11), (12), (17) and (18)?
>
> **Response:** The notation $e$ in equations (11), (12), (17) and (18) presents the the Euler constant, i.e, $e=2.718...$.
>
>
> We would like to thank the reviewer's valuable suggestion on improving the presentation, we will incorporate these in the revision. We will fix the typos and check the format of the reference list.
>
>
> **References**
>
>  [A]. Anton Rodomanov and Yurii Nesterov. Greedy quasi-Newton methods with explicit superlinear convergence. SIAM Journal on Optimization, 31(1):785–811, 2021.

---

> > ### Comment · Reviewer_YNqM · 2023-08-16
> >
> > Thank you for response. After reading other reviews and the rebuttal, I have decided to maintain my current score as it stands.

---

### Official Review · Reviewer_N1wb · 2023-07-06

**Soundness:** 2 fair
**Presentation:** 2 fair
**Contribution:** 3 good
**Rating:** 4
**Confidence:** 3

**Summary:**

The paper proposes variants of block Broyden's methods for solving nonlinear equations. Explicit convergence rates are established, and the numerical experiment validates the theoretical analysis.

**Strengths:**

The paper provides explicit convergence rates of the proposed block Broyden's methods. The theoretical analysis is sound, and the results improve some previous results related to Broyden's methods. The claims are supported by the numerical results.

**Weaknesses:**

Although this paper makes a contribution to the theoretical study of Broyden's method, there are some outstanding weaknesses that may outweigh the strengths.

1. The novelty of the proposed block Broyden's method is limited. The proposed methods are very similar to the ones given in [1] (Section 7 and Table 8.1 in [1]). The distinction should be clearly explained in the main paper.

2. The theoretical analysis is limited. Only local convergence is established. There is no discussion about the global convergence.

3. The implementation details of the algorithms are totally missing. It is likely that these algorithms cannot be efficiently applied to real-world applications. For example, each step in Algorithm 1 and Algorithm 2 needs information about the Jacobian matrix, which can incur significant overhead compared with the classical quasi-Newton methods based on secant equations (as shown in Figure 1).

4. The memory cost can be one of the biggest obstacles for the proposed methods to solving high-dimensional problems since the approximate Jacobian matrix or the approximate inverse Jacobian matrix of dimension $d^2$ needs to be maintained. However, there is no discussion about this severe issue.

5. The experimental part needs to be improved. The algorithms were only tested on a simple problem, so it is desirable to consider more different problems to make the claims more convincing. Besides, some well-known methods, e.g., the Jacobian-free Newton-Krylov method, were not compared.

Regarding to the writing, it is better to reorganize the materials to focus on the main contributions. For example, it is rather strange to analyze the convergence of the Block Broyden's update first in Section 3, which is not the algorithm proposed in this paper.

Some typos:
1. Line 29: approximate → approximates, update → updates

2. Line 150: better → better than

3. Line 209: The book [31] has no content about the Chandrasekhar H-equation.

4. Line 354: The equality is not correct.

[1] R.M. Gower and P. Richtarik. Randomized quasi-Newton updates are linearly convergent matrix inversion algorithms. SIAM J. Matrix Anal. Appl., 2017.

**Questions:**

1. The Assumption 4.1 seems to be a strong assumption. Is it possible that the nonsingularity of the matrices is guaranteed by the iterative schemes themselves?

2. The condition (10) and condition (16) require that the initial Jacobian approximation is sufficiently close to the exact Jacobian matrix. Is it a realistic assumption?

**Limitations:**

The biggest limitation of the proposed algorithms is the large memory usage and high computational cost, but there is no discussion of this issue. It is likely that these algorithms are not suitable for solving large-scale nonlinear equations in practice.

---

> ### Author Rebuttal · Authors · 2023-08-09
>
> We thank the reviewer for the detailed and helpful comments.
>
>
> >  The novelty of the proposed block Broyden's method is limited. The proposed methods are very similar to the ones given in [1] (Section 7 and Table 8.1 in [1]). The distinction should be clearly explained in the main paper.
>
> **Response:** We present comprehensive comparison to Gower and Peter's work and clearly clarify the distinction. Please refer to the global response (Section 2).
>
>
> > The theoretical analysis is limited. Only local convergence is established. There is no discussion about the global convergence.
>
> **Response:**
> Even our paper only present the local convergence rate, we think such theoretical result is significant. We improve the local superlinear convergence of nonlinear equations. Specially, the state-of-the-art result is $\mathcal{O}((1-1/d)^{t(t-1)/4})$, while ours is $\mathcal{O}((1-k/d)^{t(t-1)/4})$ (See Table 2). We also improve the local condition of the state-of-the-art method (See line 149-157).
>
> On the other hand, our method only assumes the non-degeneration of ${\bf J}({\bf x}\_\*)$ (Assumption 2.1) and the continuity of ${\bf J}({\bf x})$ along the path to ${\bf x}\_\*$ (Assumption 2.2). Hence, we think it is reasonable to focus on the local behavior around the ${\bf x}\_\*$. To the best of our knowledge, no global superlinear rate has been established for solving nonlinear equations under such mild assumptions.
>
> Though the global superlinear of quasi-Newton is established for convex optimization very recently [A], it is still unknown whether general nonlinear equation problems have similar results. We guess that it is possible to generalize the idea of [A] and use line search strategy to establish the global convergence for solving nonlinear equations. We will study it in the future.
> We are happy to incorporate more discussion based on the above response in revision.
>
> > The implementation details of the algorithms are totally missing. It is likely that ...
>
> **Response:**
> Thank you for your suggestion. We will add the implementation details of the algorithms in the revision. Actually, our algorithms **do not** require the full information of the Jacobian matrix. When updating the Jacobian estimator by the block updates (line 6 of Algorithm 1 and 2), we only need to calculate $k$ columns of the Jacobian matrix which is selected by the sampling matrix ${\bf U}_t$. Since $k\ll d$, it is cheap to obtain the partial information of the Jacobian. Using this partial information of the Jacobian can significantly improves the convergence rates compared to the classical Broyden's methods (see Table 1).
>
> The efficiency of proposed methods are also validated in our experiments part. As a result, although our methods need to calculate a bit more information per iteration, they require less running time than classical ones (see Figure 1 (d), (e), (f)).
>
> > The memory cost can be one of the biggest obstacles for the proposed methods to solving high-dimensional problems since the approximate Jacobian matrix or the approximate inverse Jacobian matrix of dimension needs to be maintained. However, there is no discussion about this severe issue.
>
> **Response:**
> Quasi-Newton methods always require $\mathcal{O}(d^2)$ space complexity to store the estimator of Jacobian (or the inverse of Jacobian) to achieve the superlinear convergence rates and efficient computation.
>
> To reduce the space complexity, limited memory quasi-Newton methods are developed for solving convex optimization (a very special case of solving nonlinear equations).
> It would be interesting to study limited-memory methods of solving nonlinear equations in the future work. We are happy to involve this discussion in revision.
>
> > The experimental part needs to be improved.
>
> **Response:** We have added experiments based on the reviewer's suggestion, please refer to the global response (Section 1).
>
>
> > The Assumption 4.1 seems to be a strong assumption. Is it possible that the nonsingularity of the matrices is guaranteed by the iterative schemes themselves?
>
> **Response:** We provide discussion on Assumption 4.1, illustrate that it is a reasonable assumption and give some potential ways to eliminate this assumption. Please refer to the global response (Section 3).
>
> >The condition (10) and condition (16) require that the initial Jacobian approximation is sufficiently close to the exact Jacobian matrix. Is it a realistic assumption?
>
> The condition that the initial Jacobian approximation is sufficiently close to the exact Jacobian matrix is standard for establishing local convergence rates of Broyden's methods ([Theorem 11.5, 31], [Theorem 1, 24] and [Theorem 4.3, 38]).
> Furthermore, the initial condition of Algorithm 1 is weaker than that of the state-of-art method [38] (line 150-161) for solving the nonlinear equations.
>
> On the other hand, we can use the matrix approximation technique in [18] to achieve a sufficient accurate initial estimator of ${\bf J}({\bf x}_0)$ (or $[{\bf J}({\bf x}_0)]^{-1}$). Since we assume that ${\bf x}_0$ is closed to ${\bf x}^*$, condition (10) (or condition (16)) is always satisfied and thus is a realistic assumption.
>
>
>
> **Response on the writing and typos:**
> We will incorporate the reviewer's valuable suggestions on improving the presentation in the revision and also fix the typos.
> * For line 209, the reference should be [20] (see Section 5.6 of [20]) rather than [31].
> * For line 354, the equality should be fixed as
>
> $$
>         {\bf J}\_\*^{\top}{\bf J}\_\* = {\bf J}({\bf x})^{\top}{\bf J}({\bf x}) + ({\bf J}\_\*^{\top}{\bf J}\_\* -{\bf J}({\bf x})^{\top}{\bf J}({\bf x})).
> $$
>
>
>
> **References**
>
>  [A]. Ruichen Jiang, Qiujiang Jin, Aryan Mokhtari. Online learning guided curvature approximation: A quasi-Newton method with global non-asymptotic superlinear convergence. Conference on Learning Theory, PMLR, 195:1962-1992, 2023.

---

> > ### Comment · Reviewer_N1wb · 2023-08-16
> >
> > Thanks for your response. While the limitation of the theory is clarified to some extent, the main issue of memory cost is not addressed. I still believe the $\mathcal{O}(d^2)$ extra memory usage is the biggest obstacle that makes these methods not applicable for solving high-dimensional problems. Gower and Richtárik's work considers the inverse of a matrix, so it is acceptable to form the matrix explicitly. However, for solving nonlinear equations considered in this manuscript, explicitly maintaining an approximation of the Jacobian or inverse Jacobian matrix in memory is very costly and should be avoided in the algorithms due to its poor scalability and potential failure when the machine resource is limited. In practice, the limited-memory Broyden's methods are often preferable to the full-memory Broyden's methods by balancing convergence and memory usage. With regard to experiments, the provided tests are small-scale, which may be not desirable since the nonlinear equations arising in realistic applications are often of high dimension. So I decided to keep the score unchanged.

---

> > > ### Author Response · Authors · 2023-08-17
> > >
> > >
> > >
> > > Thanks for your follow-up response. We politely disagree the reviewer’s comment on the limitation of memory cost. We provide the detailed rebuttal as follows.
> > >
> > >
> > >
> > > **1. Discussion on the $\mathcal{O}(d^2)$ extra memory usage**
> > >
> > > We think the $\mathcal{O}(d^2)$ extra memory usage is reasonable.
> > >
> > > * In general, the non-degenerated mapping $F:{\mathbb R}^d\to {\mathbb R}^d$ requires at least $O(d^2)$ memory to store the information of $F$, which is unavoidable to nonlinear equations.
> > > This means the memory cost of $\mathcal{O}(d^2)$ in our algorithm (or less cost in limited memory methods) does not affect the order of total memory cost.
> > >
> > > * Furthermore, even for solving the linear equations ${\mathbf A}{\mathbf x}=\mathbf{b}$
> > > with ${\mathbf A}\in{\mathbb R}^{d\times d}$ and  ${\mathbf b}\in{\mathbb R}^{d}$,  we also require $\mathcal{O}(d^2)$ memory cost to store $\mathbf{A}$.
> > > This implies solving the more difficult nonlinear equations by taking memory cost of $\mathcal{O}(d^2)$ is reasonable.
> > >
> > > **2. Discussion on inverting the matrix and solving nonlinear equations**
> > >
> > > *	As we have mentioned in above, the memory cost of $O(d^2)$ cannot be avoided to store the information of the general nonlinear mapping.
> > > This is similar to finding the inverse of $\mathbf{A} \in \mathbb{R}^{d\times d}$ requires $O(d^2)$ memory cost to store the information of $\mathbf{A}$.
> > > *	Even we focus on the problem of inverting matrix, we also improve the theoretical results of Broyden's update provided by Gower and Richtárik. Please refer to Section 2 of the global response.
> > >
> > >
> > >
> > > **3. Discussion on the limited memory Broyden's methods**
> > >
> > > To the best of our knowledge, limited memory Broyden's methods lack explicit superlinear convergence rates like ours, and there is no analysis can clearly show how to balance the convergence and memory usage of limited memory Broyden's methods.
> > >
> > > Few works have made some attempts at this point. For example, Ziani and Guyomarc’h [A] proposed limited memory Broyden's method with adaptive number of curvature pairs. However, their method need to increase the dimension of approximate
> > > to $d$ to achieve the asymptotic superlinear convergence, which means it also requires the extra $\mathcal{O}(d^2)$. Additionally,
> > > they have not provided any explicit convergence rate nor the theory on how to balance the convergence and memory. Of course, we will be happy to involve more discussion in the rebuttal phrase if the reviewer provides additional reference for the theory of limited memory methods.
> > >
> > >
> > > In summary, we think the study on limited memory Broyden's method is appropriately be left in future work and it cannot be viewed as the main weakness of our paper.
> > >
> > >
> > >
> > >
> > >
> > >
> > > **References**
> > >
> > > [A]. Mohammed Ziani, and Frédéric Guyomarc’h. An autoadaptative limited memory Broyden’s method to solve systems of nonlinear equations. Applied mathematics and computation 205.1 (2008): 202-211.

---

### Author Rebuttal · Authors · 2023-08-09

We thank all the reviewers for their detailed and helpful comments. We response to the common issues raised by the reviewers here.

### **1. Additional Experiments**
We have compared the performance of the JFNK (Jacobian-Free Newton Krylov) method with ours on H-equation in Figure 1. We observe that our method outperforms JFNK since JFNK is unstable and very sensitive to the subspace dimension $r$. In addition, JFNK does not have superlinear convergent rates.

We also test the proposed block Broyden's methods with different block size $k=\{1, 80, 500\}$ on H-equation where we set a relatively large problem dimension $N=2000$ and present the results in Figure 2.

To verify the efficiency of our methods on real-world data, we adopt quasi-Newton methods to solve the classical logistic regression:
$$
    \min\_{{\bf x}\in \mathbb{R}^d} f({\bf x}) = \frac{1}{n}\sum\_{i=1}^n \ln(1+\exp(-b\_i{\bf a}\_i^{\top}{\bf x})) + \frac{\lambda}{2}\| \|{\bf x}\|\|^2,
$$
which needs to solve the following nonlinear equations
\begin{align*}
    \lambda {\bf x} - \frac{1}{n}\sum_{i=1}^{n}\frac{\exp{(-b_i{\bf a}_i^{\top}{\bf x})}}{1+\exp(-b_i{\bf a}_i{\bf x})}\cdot b_i{\bf a}_i = {\bf 0}.
\end{align*}

We compare the proposed methods BGB and BBB with GB-Cl, BB-Cl, GB-Ra and JFNK method. We do not compare them with GB-Gr because it use greedy strategy in [38] to choose ${\bf U}\_t$,
which requires to access the full Jacobian and thus is very expensive in practice. We set the initial Jacobian estimator ${\bf B}_0=\bf I$ for all cases and validate our methods on two real world datasets a9a and w8a from the Libsvm dataset [A] and present the results in Figure 3. The results demonstrate that the proposed BGB method also outperforms the baselines significantly for the logistic regression.

We will add these additional empirical results to the revision.

### **2. Comparison to Gower and Richtárik's work**

Gower and Richtárik's work [B] mainly focus on approximately computing the inverse of a matrix. Their theoretical analysis only provide the approximation error of inverse matrix. On the other hand, our work focus on solving the general nonlinear equations, which is more challenging. We provide convergence analysis for solving nonlinear equations, which is completely not included in Gower and Richtárik's work.

Even for the matrix approximation, our theoretical results improve the results of [B].

- For the block good Broyden's update, We provide the rate of $\mathcal{O}((1-k/d)^t)$ while [Section 5, 18] (arXiv version of [B]) only provides the rate of $\mathcal{O}((1-1/d)^t)$ for the case of $k=1$.

- For the block bad Broyden's update, we provide the explicit rate of $\mathcal{O}((1-k/(d\hat{\kappa}^2))^t)$ while [Remark 7.2, B] (or [Section 8.3, 18]) only gives an implicit rate of $\mathcal{O}((1-\rho)^t)$ where $\rho = 1/\kappa_{2,F}({\bf A}{\bf S})$ can be arbitrary small.

- We evaluate our convergence property on a more general measure than that of Gower and Richtárik [18, B]. Their results hold for $\mathbb{E}[\|\|{\bf B}_t-{\bf A}\|\|_F^2]$ and $\mathbb{E}[{\|\|{\bf H}_t-{\bf A}^{-1}\|\|^2_F}]$ while our results hold for $\mathbb{E}[{\|\|{\bf C}({\bf B}_t-{\bf A})\|\|^2_F}]$ and $\mathbb{E}[{\|\|{\bf C}({\bf H}_t-{\bf A}^{-1})\|\|^2_F}]$ for any non-singular matrix ${\bf C}$.

Please also refer to Table 2 and Remark 3.3 in our paper.


### **3. Discussion on Assumption 4.1**
Assumption 4.1 is standard and widely used in analysis of Broyden's methods (See Theorem 8.2.4 of [13], Assumption 2 of [38]). We think it is reasonable to follow such common assumption.

It is possible to remove Assumption 4.1 if we moderate the initial conditions (10) (or (16)).
Intuitively, based on the non singularity of ${\bf J}\_\*$ and the point-wise smoothness of ${\bf J}(\cdot)$, we can find some local region $\Omega$ where for all ${\bf x}\in\Omega$, ${\bf J}({\bf x})$ is non-singular. When ${\bf B}\_0$ (or ${\bf H}\_0$) is sufficiently closed to ${\bf J}\_\*$ (or ${\bf J}\_\*^{-1}$), we can guarantee the non-singularity of ${\bf B}\_0$, based on Theorem 3.1 (or 3.2), we have ${\bf B}\_{t+1}$ will be closed to the ${\bf J}\_{t+1}$, and hence guarantee the non singularity of ${\bf B}\_{t+1}$ by the iterative schemes.

We thank reviewer XomP for pointing out the safeguard mechanism in [C] which can keep the Jacobians well-defined. However, the methods in [C] aim to solve minimization problem where the Jacobian (Hessian matrix) is symmetric, while the Jabobian matrix of nonlinear equations may be asymmetric. It is interesting to study how to incorporate the mechanism of [C] into solving nonlinear equations in the future.

We will involve more discussion based on above response in the revision.



**References**

[A]. Chih-Chung Chang and Chih-Jen Lin. LIBSVM: A library for support vector machines. ACM Trans-
actions on Intelligent Systems and Technology, 2(27):1–27, 2011.

[B]. Robert M. Gower and Peter Richtárik. Randomized quasi-Newton updates are linearly convergent matrix inversion algorithms. SIAM Journal on Matrix Analysis and Applications, 38(4):1380–1409, 2017.

[C]. Xiao Wang, Shiqian Ma, Donald Goldfarb, and Wei Liu. Stochastic quasi-Newton methods for nonconvex stochastic optimization. SIAM Journal on Optimization, 27(2):927-956, 2017.

---

### Decision · Program_Chairs · 2023-09-21

**Decision:**

Accept (poster)

**Comment:**

This paper studies quasi-Newton methods for solving nonlinear equations. Explicit local superlinear convergence rate is provided. One reviewer is negative towards this paper beacuse the high $O(d^2)$ memory cost. Personally, I donot think this is a big issue because this paper studies quasi-Newton methods, which require $O(d^2)$ memory cost in general. The other reviewers are positive. I suggest to accept this paper. The AC had a thorough discussion with the SAC and the SAC agreed on the decision.